# Systematic generation of biophysically detailed models for diverse cortical neuron types

Nathan W. Gouwens[1], Jim Berg[1], David Feng [1], Staci A. Sorensen[1], Hongkui Zeng[1], Michael J. Hawrylycz[1], Christof Koch [1] & Anton Arkhipov[1]

The cellular components of mammalian neocortical circuits are diverse, and capturing this diversity in computational models is challenging. Here we report an approach for generating biophysically detailed models of 170 individual neurons in the Allen Cell Types Database to link the systematic experimental characterization of cell types to the construction of cortical models. We build models from 3D morphologies and somatic electrophysiological responses measured in the same cells. Densities of active somatic conductances and additional parameters are optimized with a genetic algorithm to match electrophysiological features. We evaluate the models by applying additional stimuli and comparing model responses to experimental data. Applying this technique across a diverse set of neurons from adult mouse primary visual cortex, we verify that models preserve the distinctiveness of intrinsic properties between subsets of cells observed in experiments. The optimized models are accessible online alongside the experimental data. Code for optimization and simulation is also openly distributed.

[1] Allen Institute for Brain Science, 615 Westlake Avenue N, Seattle, WA 98109, USA. Correspondence and requests for materials should be addressed to A.A. (email: antona@alleninstitute.org)

Diverse neuronal types assemble into circuits in the mammalian neocortex. This cell type diversity has been characterized across a number of different dimensions: intrinsic physiology, morphology, connectivity, and genetic identity[1–6]. Specific subpopulations of cortical cells can be linked to particular genetic markers, and genetic tools that take advantage of these markers can provide access to these populations across a variety of experimental paradigms[7].

With this rich and expanding body of cell-type characterization, detailed computational models of neocortical circuits could serve as a framework for synthesizing a broad set of experimental data and generate hypotheses about cell-type-specific roles in the context of an active network. However, relatively few models attempt to incorporate the diversity of cellular properties observed experimentally. The largest-scale studies of this type[8] have taken the approach of generating a canonical model for each analytically defined cell type, then applying those parameters to a wide variety of morphologies to generate variations in intrinsic properties. An alternative to this approach would be to fit many individual cells that have each been characterized experimentally, then populate a network model by drawing from this large model library, without necessarily defining cell types in advance.

One challenge in creating a large library of individual cell models is that the characterization of different cell types is frequently done by different laboratories under different conditions. It is difficult for modelers to gather a set of data across many cells that facilitates the generation of models in a consistent way. Since the balance of active conductances that governs a neuron's intrinsic electrophysiological behavior is finely tuned, it is difficult and time-consuming to optimize the combination of model parameters that accurately reproduces the target neuron's activity.

To address this challenge, several studies have described automated parameter fitting approaches with multicompartment conductance-based models[9–17]. These approaches include several different (though sometimes overlapping) optimization methods (e.g., genetic algorithms, simulated annealing, downhill simplex) and target objective functions (e.g., direct fitting of voltage traces, feature-based comparisons, phase plane comparisons). In addition, many of the more recent studies have released open-source code so that others may use similar methods. However, these studies typically demonstrate their methods on a limited number of experimental examples, which often represent a relatively specific cell type, such as cortical layer 5 pyramidal neurons. Therefore, it is not clear how easily the methods can be transferred to novel cell types. In addition, while the parameter fitting is automated with these methods, setting up the methods to apply them to many cells can require additional manually executed steps.

Here we present an approach for automatic optimization of biophysically detailed neuronal models and a set of 170 models generated from a high-throughput experimental data pipeline (the Allen Cell Types Database[18]). These models are systematically generated for a wide variety of cell types based on experimental data collected via a highly standardized protocol from the primary visual cortex of the adult mouse. The models are based on individual recorded cells that in most cases were labeled by a specific transgenic driver line, and the locations of the recorded cells in the brain were mapped to a standard three-dimensional (3D) reference space (the Allen Mouse Common Coordinate Framework[19]). We show that this optimization procedure generates models that reproduce essential features of the electrophysiological properties of the original cells and generalize across a range of stimulus types that were applied in experiments. This flexible analysis and optimization approach is publicly available as open-source code, which has the advantages of being relatively concise, extendable, and based upon open-source, well-supported libraries. Additionally, we use classification methods to demonstrate that the model set largely preserves the distinctiveness across cell types found in the original data. Together, this model set provides the fundamental components for larger models of neocortical networks.

## Results

**Model fitting procedure.** We developed an automated analysis and optimization procedure to allow for the systematic generation of models from a high-throughput experimental pipeline (Fig. 1). We implemented our methods in Python via a code base that drew on open-source projects for improved ease of use and better maintenance. These software projects include the Allen Software Development Kit (Allen SDK)[20] for electrophysiological feature analysis and simulation control and the Distributed Evolutionary Algorithms in Python (DEAP) library[21] for flexible optimization via genetic algorithm. The integration with these packages enabled the flexible use of different analytic and optimization routines. In addition, different styles of fit (e.g., different combinations of conductances, different feature sets) could be quickly implemented via configuration files using the widespread JavaScript Object Notation file format. Sample code illustrating the use of these libraries is shown in Fig. 1b, and the code implementing these procedures is publicly available.

Using this standardized model optimization procedure, we generated biophysically detailed models for 170 cells from the Allen Cell Types Database[18]. Figure 1a depicts the basic workflow for model production. We based the models on whole-cell recordings made from adult (P45 to P70) C57BL/6J mouse visual cortical slices using standardized conditions and protocols. In many cases, the neurons targeted expressed the tdTomato reporter under the control of a variety of Cre-based transgenic driver lines. The transgenic lines were selected to target three major interneuron types: those expressing parvalbumin (*Pvalb*), somatostatin (*Sst*), and the serotonin receptor 5HT3a (*Htr3a*). Other lines labeled pyramidal neurons in specific layers (e.g., *Nr5a1*, *Ntsr1*, *Rbp4*). Responses to a fixed electrophysiological protocol were recorded for each neuron, and neurons were filled with biocytin during the recording. Dendritic morphologies were reconstructed from the biocytin fills when possible.

From this data set, we first built an entirely passive model using the somatic and dendritic morphological reconstruction to estimate the passive parameters (see Fig. 2). Next, we selected the cell's responses to a single suprathreshold current step stimulus and calculated the values of a set of electrophysiological features (see Methods section for definitions) from those responses. These included features characterizing the action potential shape (e.g., peak, width, trough depth) and firing pattern (e.g., average rate, adaptation index, latency to first spike). For approximately one-third of the cells, multiple repeats of a given stimulus amplitude were available. For these cells, we also measured the intrinsic variability of these features from repeat to repeat.

Next, we added active conductances to the soma of the model cell and used a genetic algorithm to optimize the densities of these conductances so that the model features would match those of the original experiment. We assessed the ability of the model to match the experimental features by calculating the error of each electrophysiological feature as an absolute *z*-score and averaging them together to produce a single objective for the optimization procedure ("average training error"). We found that this single-objective optimization approach converged more rapidly to acceptable solutions than the multi-objective approaches we evaluated.

The set of model parameters with the lowest average training error were added to the database of models if it was below the

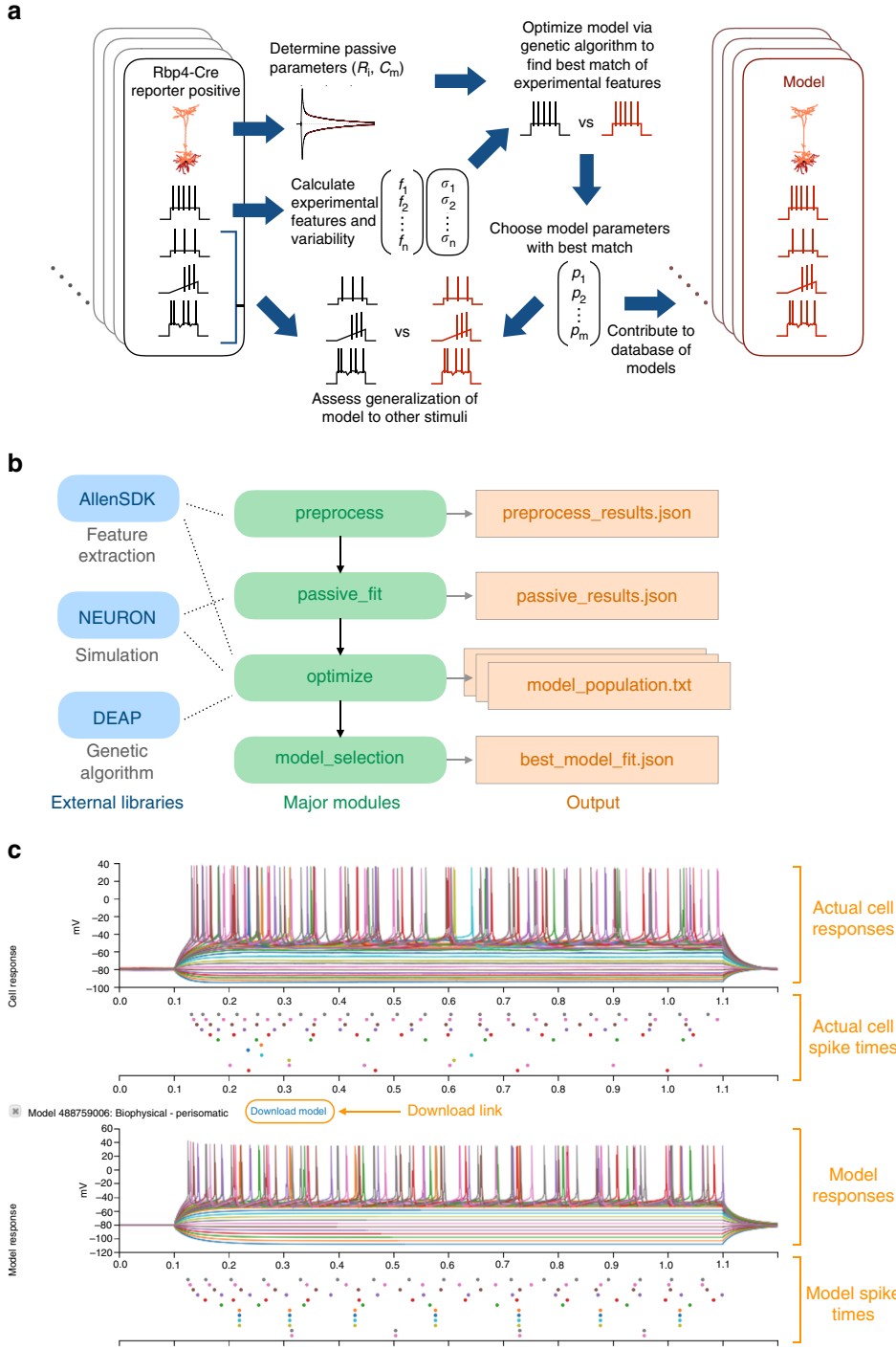

**Fig. 1** Overview of model fitting procedure and resources. **a** Schematic showing the steps involved in using the standardized data from individual neurons in the Allen Cell Types Database to generate optimized models that reproduce aspects of firing patterns. **b** Schematic illustrating the structure of the code, including external open source libraries (blue), major modules for model optimization (green), and key output files (orange). Note that the optimization module specifically is written to be run automatically in a parallel fashion. **c** Screenshot of published models alongside the experimental data from the Allen Cell Types Database web application. Annotations to the screenshot are in orange. Note that multiple sweeps of 1 s long hyperpolarizing and depolarizing current steps are overlaid in different colors

inclusion threshold (see Methods section). Each model was also compared to the experimental data by applying all the stimuli used in the original experiment. These methods were used to generate models for a wide variety of cortical neurons with diverse morphologies and intrinsic properties.

Our modeling results have been made available alongside the original data in the Allen Cell Types Database (Fig. 1c), allowing

others to compare easily between models and experiments. Users of the website can also download the models to run themselves; the ability to execute models quickly without much additional configuration is supported by the Allen SDK package.

**Staged model fitting**. As mentioned above, we implemented a staged approach to fitting the neurons. To start, we directly fit the

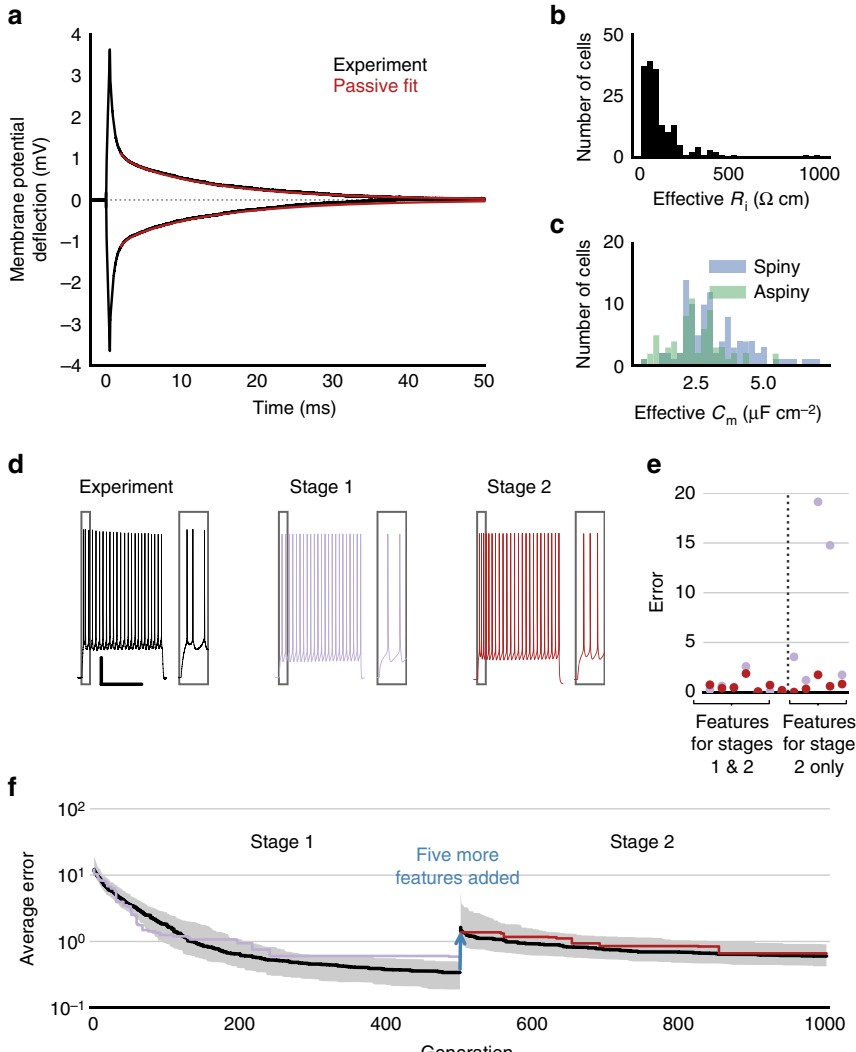

**Fig. 2** Staged fitting approach. **a** Direct fitting of the return to baseline after a de- and hyperpolarizing current step (0.5 ms, ±200 pA) with an all-passive model. **b** Distribution of effective specific intracellular resistivity values ($R_i$) across all models ($n = 170$) determined by passive fitting. **c** Distribution of effective specific membrane capacitance values ($C_m$). **d** Voltage traces from an example cell showing the experimental data used for optimization (left) and the model responses after the first (middle) and second (right) stages of optimization; scale bar, 20 mV (vertical), 1 s (horizontal). Boxes highlight the initial spikes. **e** Individual feature error values (defined as absolute z-scores compared to experimental values) after the first stage (purple) and second stage (red). Note that the first stage values to the right of the dashed line are high because those features were not part of the error function in the first optimization stage. **f** Average error (absolute z-score) across all features for the best model in a given generation. Black shows the median values across all models, and the gray region shows the interquartile range. Purple and red traces show the specific error trajectory for the cell in **d** and **e**. Note that the jump at generation 500 occurs because five new features are introduced, and the average error at the end of stage 2 is higher than at the end of stage 1 because it is more difficult to reduce the error across 12, rather than 7, features

responses to brief (0.5 ms) 200 pA subthreshold current steps using a morphologically detailed model with uniform specific passive parameters (Fig. 2a). That is, the specific membrane capacitance, specific membrane resistance, and specific intracellular resistivity were varied to best match the decay of the membrane potential back to baseline after the step. This procedure minimizes the effects of artifacts from the recording pipette[22, 23].

The passive parameters estimated by this direct-fitting procedure varied across the population of models (Fig. 2b, c). The $R_i$ values were typically below the oft-reported value of 100 $\Omega$ cm for mammalian neurons, while values of $C_m$ were typically larger than 1 $\mu$F cm$^{-2}$. On average, spiny neurons had larger $C_m$ estimates than aspiny neurons (Fig. 2c). These differences could have arisen from systematic biases of the morphological reconstructions, such as underrepresentation of dendritic

diameters, shrinkage due to tissue processing, and a lack of explicit representation of dendritic spines. In addition, excluding the axon from the passive fits (which was necessary as axonal reconstructions were not available for most cells) can lead to overestimates of $C_m$ values[23]. An analysis of these factors (Supplementary Fig. 1) showed that the fitting procedure returns values much closer to the expected $C_m$ of 1 $\mu$F cm$^{-2}$ when corrections to the dendritic morphologies were made and the axon was included; in particular, the addition of the axon had the largest effect on the $C_m$ estimate. In light of this, it is important to exercise caution in interpreting the parameter estimates reported in Fig. 2b, c. Rather than providing precise representations of passive parameters for the actual cells, fitting the dendritic morphologies alone instead furnished effective parameters enabling the model to mimic the electrophysiological properties of the cell. With this approach, we were able to achieve good fits

to passive responses (Fig. 2a) to generate a broad library of individual cell models, despite incomplete morphological information. Clearly, fitting the passive properties using both dendritic and axonal morphologies is the more appropriate approach for applications where knowing the exact (rather than an effective) value of parameters such as $C_m$ is important, such as investigations of systematic differences in $C_m$ like those recently reported between mouse and human cortical neurons[24].

Next, we placed active conductances at the soma of the cell and optimized the parameters of the active model. The free parameters adjusted during optimization were the densities of each active and passive conductance, as well as two parameters that affected how intracellular $Ca^{2+}$ was handled in a submembrane shell (see Methods section). The passive parameters $R_i$ and $C_m$ were fixed during optimization to values determined by the earlier direct passive fitting. $R_i$ was set to be uniform in all compartments, while $C_m$ was set to a higher value in dendrites for spiny neurons (see Methods section). To judge the goodness of fit, we calculated a set of features from the responses to a single amplitude step current injection, which served as the training data for model optimization. The error in a given feature was calculated as the absolute value of its $z$-score. The standard deviation used for the $z$-score calculation was the experimental standard deviation when repeated experimental responses at the same stimulus amplitude were available or predetermined tolerances when only a single response was recorded (see Methods section and Supplementary Table 1). The objective for the optimization procedure was the average of the absolute $z$-scores across all target features.

We found that including all target features from the start led to poor model convergence in some cases. Therefore, we first fit the models with a limited set of features (Stage 1) that represented basic aspects of the cell's firing pattern, such as action potential shape and firing rate. After that, we took the best models from that stage and continued fitting on all target features (Stage 2) (Fig. 2d–f), which included more subtle aspects of firing such as time to the first spike and spike frequency adaptation. In the example shown in Fig. 2d, the best-fit model after Stage 1 exhibits a slower initial firing pattern compared to the original experiment, even though other properties such as the action potential height and average rate are closely matched. Adding features such as the duration of the first interspike interval (ISI), the latency to the first spike, and the adaptation index to the second stage of fitting allows the optimization routine to find a best-fit model that better reproduces the initial firing of the cell.

**Issues encountered during fitting.** We originally fit neurons using a set of conductances based on those employed by Hay et al. (2011)[13] for fits of layer 5 pyramidal neurons (set A)[13, 25–34]. However, we found that optimizations of cells that had narrow spike widths (most frequently seen in interneurons) frequently were unable to reproduce those target widths (Fig. 3a, b).

We reasoned that this failure could be due to the inability of the kinetics of the underlying conductances to support rapid action potentials. Therefore, we attempted alternative optimizations in which the sodium conductance and several potassium conductances were replaced with other conductance models (set B, Fig. 3d) based on additional data in the literature[35–38]. We found that these substitutions led to improved fits of the narrow spike widths (Fig. 3b, c). In the example model fit with set B (Fig. 3b), the sodium current inactivated more completely by the time of the peak of the action potential, which may enable the optimizer to select a different mix of potassium conductances that more accurately reproduces the trajectory of the action potential downstroke. We therefore typically fit cells with the narrowest

action potentials (<0.5 ms width) with conductance set B only ($n = 60$), cells with the widest action potentials (>1 ms width) with conductance set A only ($n = 65$), and cells with intermediate widths with both sets (Fig. 3c). For the intermediate cells ($n = 45$), we compared the two fits and selected the model with the lower overall training error (set A: $n = 23$, set B: $n = 22$).

Another issue we encountered was that a sizeable fraction of optimized models reproduced the training stimulus well but then displayed substantial depolarization block when higher-amplitude current stimuli were delivered (Fig. 4a). This also was reflected by a strong falloff in the $f$–$I$ curve for those models, which the actual cells did not exhibit (Fig. 4b).

As mentioned above, the genetic algorithm-based optimization method we used produces many sets of model parameters at the end of the procedure, rather than a single optimal set. We tested these other models produced in the same optimization run and found that many of them did not exhibit the depolarization block observed in the model with the lowest average training error (Fig. 4c). Therefore, to improve the chances that these models would be used in later generations of the optimization procedure, we introduced an automatic check for depolarization block that strongly penalized models which displayed the block. Namely, in addition to simulating the training step for each model, the model-fitting procedure simulated the response to an additional current step that matched or exceeded the highest current amplitude applied to the original cell during the experiment. This additional response was evaluated for a failure to repolarize during or after the current step. If that failure was detected, an additional penalty was automatically applied to the error function. Since the check required additional simulations during optimization and therefore extended the overall execution time, the check was only applied during the second stage of optimization. Implementation of the check resulted in models that still fit the training stimulus well (Fig. 4a) but did not fail to sustain firing at higher current steps (Fig. 4a, b).

**Generation of a diverse model set.** Models generated by this procedure reproduced the firing patterns upon which they were trained across a range of cells recorded, from regular-spiking pyramidal cells to fast-spiking interneurons (Fig. 5a). Average training errors (each an average of absolute $z$-scores across all features) fell between 0.16 and 3.01 with a median of 0.59 and an interquartile range of 0.42–0.89 (Fig. 5b). Models with low average training errors were successfully generated across all transgenic lines tested, though models with high average errors were also observed across most transgenic lines.

While the average training errors observed were frequently low (absolute $z$-score <1), certain features were more easily fit well than others (Fig. 5c). The model features that had the highest errors at the end of the procedure were most frequently the membrane potential during the slow trough and the action potential width, while the features that were least likely to have the highest errors were the baseline membrane potential, the average firing rate, and the average ISI (the latter two, of course, are highly correlated). These differences could reflect a variety of influences, ranging from the intrinsic abilities of the active conductances used to reproduce the repolarization phase of the action potential to the relative weights placed on different features by the optimization routine due to the observed experimental variability and tolerance levels selected.

**Model generalization.** Next, we assessed how the models performed on stimuli upon which they were not trained. We focused on responses to noisy current injection, slow current ramps, and responses to a series of depolarizing current steps. These diverse

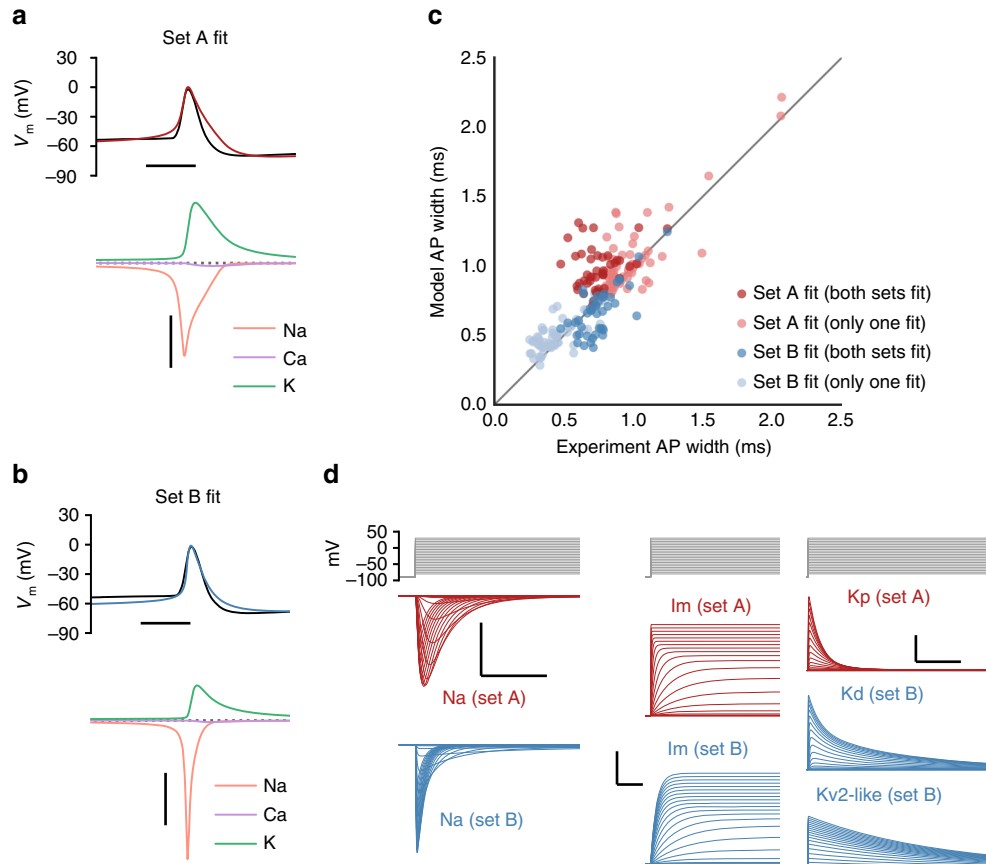

**Fig. 3** Modeling narrower spike widths. **a** Voltage traces of an individual action potential from the experiment (black) and model fit using conductance set A (red); scale bar, 2 ms. The major currents flowing during the action potential are shown below; scale bar, 2 nA. **b** Voltage traces and currents during an individual action potential from the experiment (black) and model fit using conductance set B (blue); scale bar, 2 ms (horizontal), 2 nA (vertical). **c** Comparison of model and experimental action potential widths across all models. Red dots are models fit with conductance set A; blue dots are models fit with conductance set B. The darker dots represent cases when both types of fits were performed on the same cell, which was typically done at intermediate spike widths. **d** Voltage-clamp measurements of model currents in response to a range of holding potentials (–80 mV to +40 mV) from –90 mV. Currents that are from conductance set A are shown in red. Currents from the set B sodium conductance (left, scale bar: 2 nA vertical, 2 ms horizontal), M-type potassium conductance (middle, scale bar: 5 nA vertical, 50 ms horizontal), and slow potassium conductances (right, scale bar: 3 nA vertical, 500 ms horizontal) are shown in blue. Note that a single slow potassium current in set A was replaced by two potassium conductances in set B

stimuli were consistently applied during the original experiments, allowing us to make direct comparisons between the model and experimental data.

Models performed relatively well on noisy current injection (Fig. 6a–c) in terms of the timing of action potentials evoked by the stimulus. The pink noise stimulus was designed to mimic the statistics of in vivo synaptic input and has been used to fit generalized leaky integrate-and-fire models of individual neurons in the database[39]. Here we quantified the performance of our models on this stimulus by comparing the variance of spike times explained by the model to the variance explained by the trial average of the experimental data ("explained variance," see Methods section). Models with lower average training errors exhibited better performance on the noise on average (Fig. 6c); however, there were notable exceptions in both directions (i.e., high feature errors but high explained variance ratios, and low feature errors but low explained variance ratios).

We found that our models also could generalize to ramp stimuli (Fig. 6d–f). As the ramp stimuli in the experiments were terminated after eliciting a few action potentials to prevent overstimulating the neurons, our experimental data set for this stimulus type did not have consistent trains of action potentials to analyze. Therefore, we focused our comparisons on properties of

the first action potential evoked by the slow ramp stimulus. The distribution of the average difference in latency to first spike between the models and experiments was centered near zero; however, there was a broad range in latency differences across cells (Fig. 6e). The height of the action potential on the ramp stimulus was similar to the experiment but displayed a consistent bias toward shorter action potentials in the models (Fig. 6f).

Model f–I curves were also generally similar to those of the original cells they were based on (Fig. 6g, h) in terms of rheobase (Fig. 6i) and slope (Fig. 6j). Notably, however, the optimized models for fast-spiking neurons tended to exhibit an f–I curve slope that was not as steep as the slope measured experimentally (Fig. 6g, i). Since the models were optimized to match the high-firing rate observed on the training step, models that exhibited a lower-than-expected slope correspondingly had a lower-than-expected rheobase (Fig. 6g–j). Aside from this class of neurons, models exhibited good agreement with the experiment on both the slope and the rheobase (Fig. 6i, j).

**Models and cell-type characterization.** Given that we generated a set of models that encompass a variety of transgenic line-based labels, layer locations, and firing patterns, we investigated the

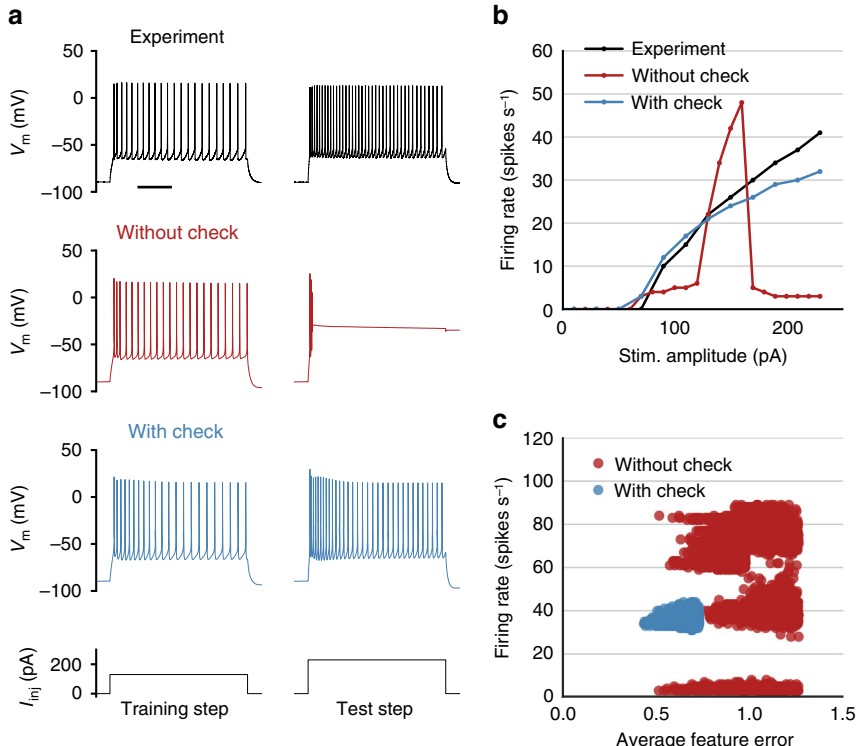

**Fig. 4** Addressing depolarization block during optimization. The same example cell is shown in **a**–**c**. **a** Voltage responses to step current injections of two amplitudes (left: 130 pA, right: 230 pA) from the original experiment (top), a best-fit model without an on-line penalty for depolarization block (middle), and a best-fit model with an on-line penalty for depolarization block (bottom); scale bar, 250 ms. **b** Firing rates measured in the experiment (black), by the model without a penalty (red), and by the model with a penalty (blue) in response to a range of step current injections. Note the drop in firing rate in the red trace after 170 pA due to the onset of depolarization block. **c** Firing rates elicited by a high (240 pA) current step from all 1200 models generated by the genetic algorithm at the end of optimization for the example cell compared to the average feature error of each model. Red points are the models generated without a depolarization block penalty, and blue points are models generated with a depolarization block penalty

relationship between the models and cell-type classification based on intrinsic electrophysiological properties. Since we modeled the cells with a similar set of conductances, it is possible that the models could be more similar than the actual cells (which presumably express a variety of different conductances across cell types). As a first step, we separated the cells that expressed a transgenically driven reporter and their associated models into categories based on their transgenic line. We selected cells from three major interneuron classes based on the markers *Pvalb*, *Sst*, and *Htr3a*. In addition, we grouped the excitatory-dominant driver lines into a single "pyramidal" class (Pyr).

Next, we tested whether the original 12 experimental features used to train the models could be used to predict membership of these four classes with a support vector machine classifier. The transgenic line-based classes could be predicted with the experimental features with 79% accuracy (Fig. 7a, left). The pyramidal cell class was the most reliably predicted (91%), followed by the *Pvalb* (80%), *Sst* (59%), and *Htr3a* (50%) classes. This could not only reflect inaccuracies in the classifier but also could reflect overlapping electrophysiological characteristics between the different groups of interneurons.

We then used the electrophysiological features of the optimized models for classification (Fig. 7a, middle left). We observed broadly similar results (81% overall accuracy) to the classification with experimental features, with rates of identification of the Pyr (88% accuracy), *Pvalb* (96%), *Sst* (59%), and *Htr3a* (58%) classes similar to those obtained with experimental features. The overall similarity in classification performance suggests that the models preserve much of the distinctiveness in firing properties between these groups of cells.

We then tested whether performing classification using the model parameters (i.e., the conductance density values) rather than the model features (i.e. action potential peak, width, etc.) could also reliably discriminate between these transgenic line-based groups of cells (Fig. 7a, middle right). A classifier trained on those features performed somewhat less well than the others (64% overall accuracy). The primary difference was that interneurons were more frequently confused with each other, lowering each of their classification accuracies (*Htr3a* 47%, *Sst* 29%, *Pvalb* 64%). However, since the same set of conductance densities would produce different firing patterns when inserted into different morphologies, it is possible that better discrimination could be achieved if morphology-related values were made available to the classifier. We found that when several of these values (total capacitance, the soma-to-dendrite capacitance ratio, and the average attenuation from soma to dendritic tip) were added to the set of model parameters for classification, performance improved somewhat (Fig. 7a, right) to 70%.

The transgenic line-based groups may still encompass a wide variety of overlapping electrophysiological types, which would limit the best-case performance of these types of classifiers. Therefore, we next divided the models into different classes based on an unsupervised clustering of electrophysiological features measured with a variety of stimulus protocols[39] on a larger set of cells ($n = 645$) in the Allen Cell Types Database. The cells we modeled here fell into the 16 clusters identified by that broader analysis; however, we only assessed the 10 clusters that contained at least 5 modeled cells. We then built classifiers using the experimental features, model features, and model parameters as with the transgenic line-based groups (Fig. 7b) to predict these clusters.

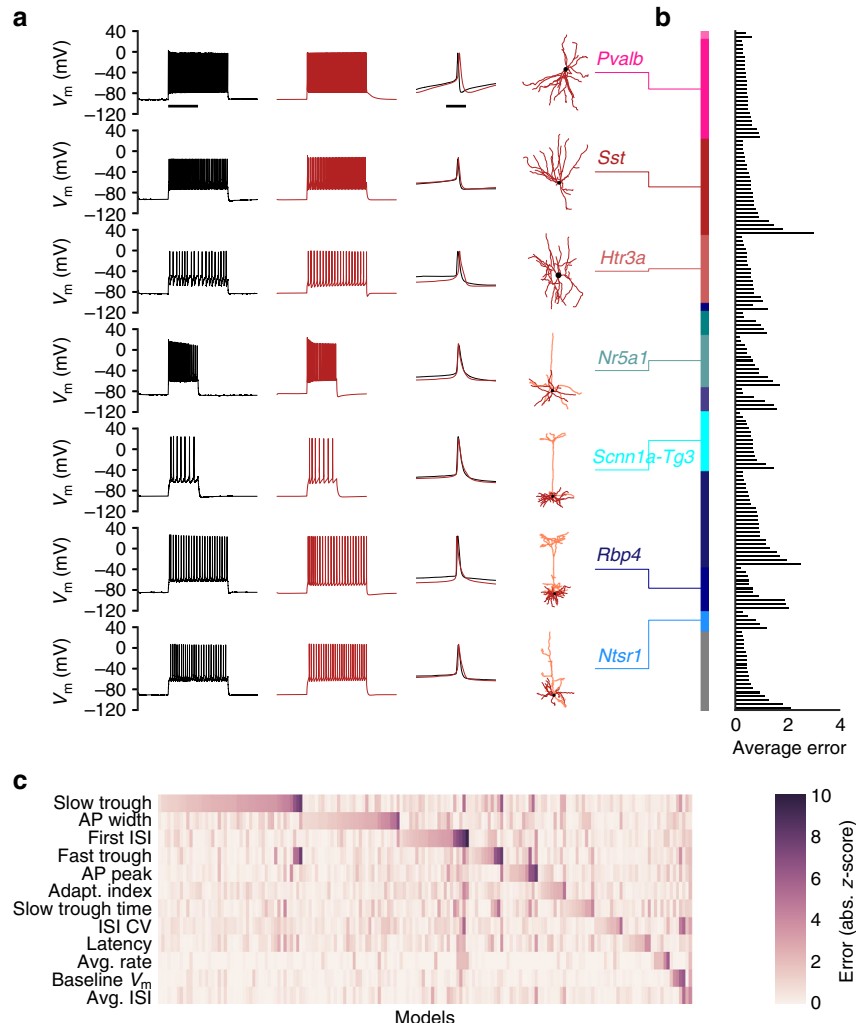

**Fig. 5** Example models and summary of training errors. **a** Voltage traces showing the experimental responses (black, scale bar: 1 s) and model responses (red), including a comparison of individual action potentials (scale bar: 5 ms), for seven neurons from a variety of transgenic lines. The dendritic morphologies for each cell are also shown. **b** Average training errors for all 170 models. Models are organized by transgenic line (color bar; groups shown from top to bottom are Gad2, Pvalb, Sst, Htr3a, Slc17a1, Cux2, Nr5a1, Scnn1a-Tg2, Scnn1a-Tg3, Rorb, Rbp4, Ntsr1, and unlabeled) and then in ascending order of average training error. **c** Training errors for specific features for each of 170 models. Errors are reported as absolute z-scores. Models are sorted by the feature that had the highest remaining error

We found that the 10 clusters could be discriminated fairly accurately (53%) based on the 12 experimental features used to train the models. Classification using the model features exhibited similar performance (52%), again suggesting that the models preserve distinctiveness across these more granular classes. To support this, we also found that the distributions of experimental and model features by class were similar (Supplementary Figs. 3 and 4) and that classifiers trained on experimental features could predict classes using model features, as well (Supplementary Fig. 5). Using the model parameters, accuracy fell to 24%. However, inclusion of morphology-related features increased the accuracy to 44%—a large improvement but not as accurate as classification based on features.

Several factors may contribute to this result. It is possible that certain model parameters have only a limited effect on the firing properties of the cell, which could in effect add noise and make the classification task more difficult. While we did perform feature selection by recursive feature elimination and cross-validated selection when building the classifier, this may not have been able to fully counter those types of effects. Also, more errors occurred by confusing similar classes with each other rather than

by confusing, for example, fast-spiking classes with regular-spiking classes. Overall, despite a lack of discrimination with model parameters alone, the generated set of models do appear to preserve electrophysiological distinctiveness found in the experimental data.

## Discussion
The mammalian neocortex exhibits great diversity in its component cells across multiple dimensions, which experimentalists continue to characterize in detail. However, this diversity presents challenges for modeling studies of cortical circuitry, in part because detailed biophysical models of networks rely on well-tuned models of individual cells. To address this issue, we implemented an analysis and optimization workflow, applied it across many cells, and produced models that reflect electrophysiological activity of the original recorded neurons. We made use of an electrophysiological and morphological data set collected in a consistent way across a wide set of cortical neurons to produce the 170 single-cell models presented here. We also have made the workflow code we developed openly available to allow others to generate new models in an automated fashion.

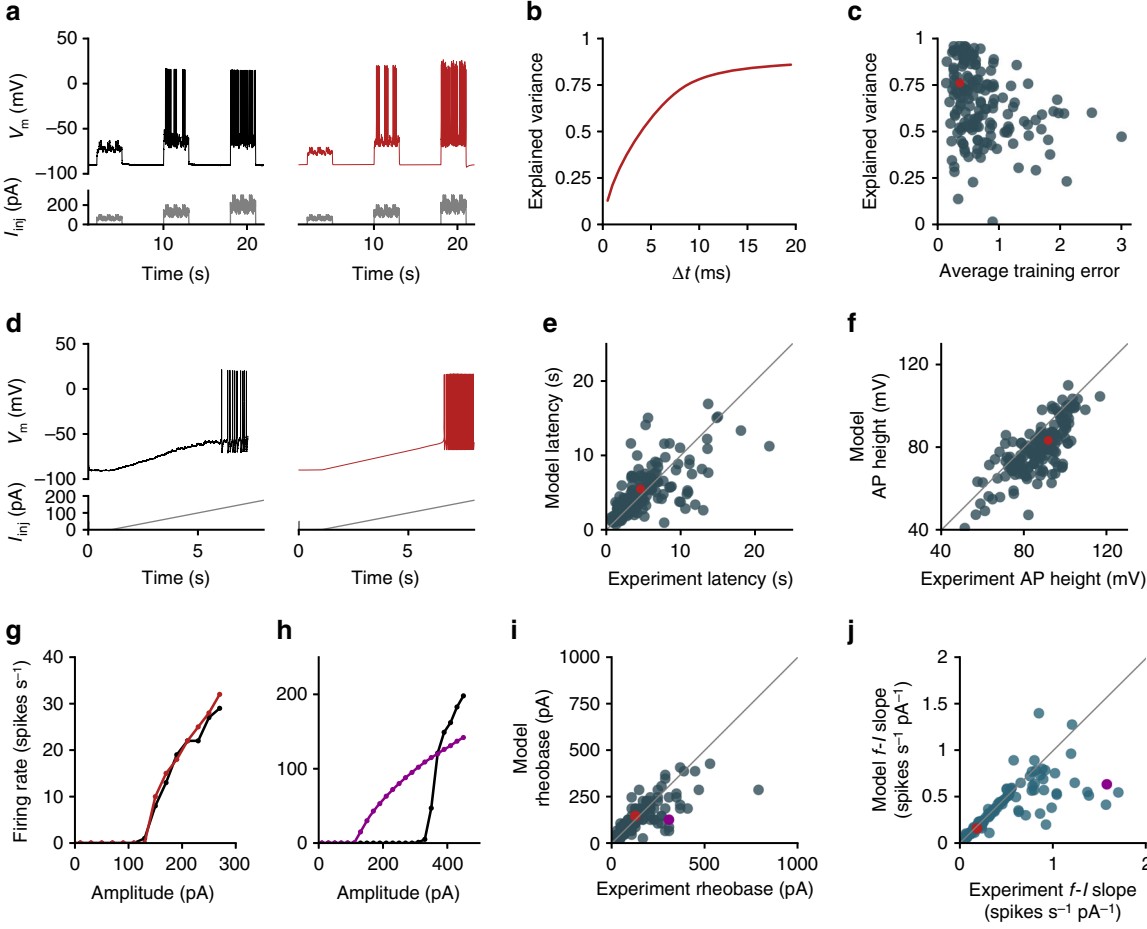

**Fig. 6** Model generalization. **a** Responses to noisy current injection from the experimental data (black, left) and model (red, right). One experimental example from several repeated presentations of the stimulus is shown. **b** Explained variance of spike times as a function of the time window ($\Delta t$) resolution for the cell in **a**. **c** Comparison between the explained variance with a time window $\Delta t = 10$ ms and the average training error. **d** Responses to a slow ramp current injection (25 pA s$^{-1}$) from the experimental data (black, left) and model (red, right). One experimental example from several repeated presentations of the stimulus is shown. **e** Comparison between the latency to first spike during the ramp stimulus for the model and experimental data. **f** Comparison between the height of the first action potential during the ramp stimulus for the model and experimental data. **g** Average firing rates evoked by a range of current steps in the experiment (black) and by the model (red) for an example cell. Data from **a**, **b**, **d**, and **g** are from the same pyramidal (*Nr5a1*-labeled) cell, also indicated by red dots in **c**, **e**, and **f**. **h** Average firing rates evoked by a range of current steps in the experiment (black) and by the model (purple) for a fast-spiking (*Pvalb*-labeled) interneuron. **i** Comparison of rheobases for the model and experimental data assessed by current steps at 20 pA increments. **j** Comparison of slopes for the model and experimental data measured by linear fit of the non-zero part of the *f–I* curve. Data from examples in **g** and **h** are shown as red and purple dots, respectively, in **i**, **j**

The models we have generated do not necessarily reflect a unique combination of parameters that results in the best fit to the experimental data. The genetic algorithm approach used to search the parameter space is not guaranteed to find a global minimum. In addition, different combinations of ion channels can produce similar firing activity[40]. This may contribute to our finding that model parameters (i.e., the specific conductance densities for a particular cell) were less useful at discriminating between major subpopulations of cells than the electrophysiological features exhibited by the model (Fig. 7). Interestingly, when neurons are fit with simpler generalized leaky integrate-and-fire models that guarantee unique parameter solutions, clustering on those model parameters produce classes that are relatively consistent with different electrophysiological types and transgenic lines[39]. Despite not representing unique solutions, though, our models generalize across a range of different stimulus types (Fig. 6) and preserve some distinctiveness across these subpopulations (Fig. 7), suggesting that they could be appropriate for introducing cell-type heterogeneity into large-scale models of cortical circuits.

We do note several limitations of our models in terms of reproducing behaviors observed in the original cells. For example, the models do not exhibit a rapid onset ("kink") at the start of the action potential, which is due to axial current entering the soma after action potential initiation in the axon[41–43]. We investigated model fits that included axonal active conductances, which in some cases did demonstrate a rapid action potential onset. However, the introduction of these additional parameters frequently prevented the algorithm from converging to an acceptable solution for many cells, given the computing resources available to us. Still, refinements or constraints applied to the optimization procedure may allow models with axonal conductances to be reliably generated in the future. Along similar lines, the models only have active conductances placed at the soma. While this reduction in complexity reduces the computational resources required to optimize and simulate the models, it will also affect how synaptic input is integrated in the models. Addressing these issues will require future work, including comparison with models built from the same experimental data that have active conductances in

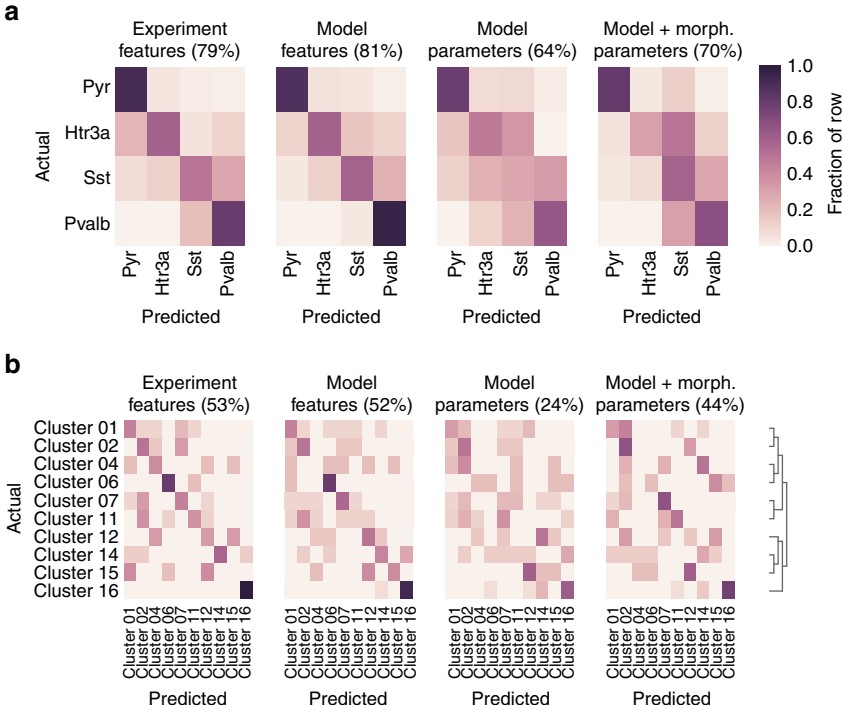

**Fig. 7** Prediction of cell classes. **a** Prediction of transgenic line-defined cell classes ($n = 146$ cells). Note that all excitatory-dominant transgenic lines were combined into the "Pyr" category. **b** Prediction of cell classes generated by unsupervised clustering that used electrophysiological features from a broader set of cells based on the clustering from ref. [39]; only clusters containing at least 5 cells from the set of models constructed here are shown ($n = 89$ cells). Note that the referenced analysis identified 16 clusters; the cluster names shown here are consistent with that study, and clusters with too few cells in our study (i.e., clusters 3, 5, 8, 9, 10, and 13) are omitted. The tree at the far right shows the relative hierarchical relationships between the clusters. Predictions were made by a supervised classifier that used either the features of the experimental data that models were trained on (left), features of the optimized models (middle left), the parameters of the optimized models (middle right), or parameters of the optimized models as well as morphology-related values (right). Each entry within a heatmap shows the fraction of cells/models in a given row assigned to a particular column

dendritic compartments also publicly available online (see refs. [44] and [45]).

Another limitation is evident with the models of fast-spiking cells. In many cases, the generated models could replicate the fast dynamics of individual spikes but not the steep $f$–$I$ curve associated with that type of firing. The relatively shallow $f$–$I$ curve of the model also led to the rheobase of the models being lower than found in the experimental data; otherwise, the models would not reach a high enough firing frequency at the stimulus amplitude used to train the models. While some models of fast-spiking cells did not exhibit this discrepancy, large differences in the $f$–$I$ curve were most common among this class of cells. This mismatch could be due to several factors, including our use of a set of active conductances originally employed for fits of excitatory pyramid neurons[13]. Our modifications to achieve narrow action potential widths (Fig. 3) improved the overall fits of fast-spiking neurons, but further modifications may be necessary to improve this aspect of the models' firing activity. Finally, there may be unmeasured complex firing features[46] possessed by the original cells that our models do not capture. Since our approach involves creating models of individual cells, our evaluations of the generalization of the models are constrained to the stimuli used in the original experiments. Additional experimental studies may reveal aspects of our model generation approach that require further refinement.

Several publicly available tools for multicompartment, conductance-based single-cell models have been recently described[11, 14–16]. These tools have been optimized for different trade-offs, such as graphical user interfaces vs programmatic interfaces, choices of objective functions and optimization

methods, and integration with other analysis tools. The appropriate choice for researchers will depend on their experimental and technical circumstances. For the method we developed here, we emphasized the use of other flexible open-source packages, robustness across a diverse set of cell types, and automation of its use across a large experimental data set.

For example, we have employed a single-objective optimization method, as opposed to a multiobjective technique, since through initial testing we found more rapid convergence on acceptable solutions with this method compared to others. This was similar to another study that found that combining objectives, even when using a multiobjective method, can reduce computational resources and improve convergence[13]. Also, our intent was to produce a set of representative models that corresponded to individual experimentally characterized cells, so if we were to use a multiobjective method, we would still have employed model selection criteria to identify a representative example that would likely resemble how we combined objectives in this study. It is also important to note that our use of an open-source evolutionary algorithm library[21] means that it is a straightforward code change to use alternative algorithms implemented in that library, including several multiobjective options.

Our goal was to produce a set of models spanning many cortical cell types to support large-scale modeling studies. To facilitate the use of these models in future simulations, they are publicly available online alongside the experimental data upon which they are based in the Allen Cell Types Database, as well as through other established community data-sharing repositories[47, 48]. Python code is provided in the Allen SDK to run the models with limited set-up required from the user.

Building network models from single-cell models that can be closely associated with specific experimental tools and metadata (e.g., transgenic line, cortical location) will clarify comparisons with in vivo experiments that employ those same identifiers.

## Methods

**Experimental data collection.** Models were based on data collected as part of the Allen Cell Types Database[18]. Electrophysiological data were recorded by in vitro whole-cell recording from neurons of adult mouse visual cortex[49]. In brief, slices were prepared from P45–P70 mice that expressed tdTomato via the *Ai14* reporter line[50] crossed with a specific Cre driver line[51]. Whole-cell recordings were typically made from tdTomato-positive neurons (though in some cases tdTomato-negative cells were specifically targeted), and a standard electrophysiogical protocol was applied consisting of current steps, ramps, and pink noise at various durations and amplitudes. Recordings were made at 34 °C in an external solution consisting of 2 mM calcium chloride (dehydrate), 12.5 mM D-glucose, 1 mM magnesium sulfate, 1.25 mM sodium phosphate monobasic monohydrate, 2.5 mM potassium chloride, 26 mM sodium bicarbonate, 126 mM sodium chloride, 1 mM kynurenic acid, and 0.1 mM picrotoxin. Recording pipettes (borosilicate, 3–7 MΩ) were filled with an internal solution consisting of 126 mM potassium gluconate, 10 mM HEPES, 0.3 mM ethylene glycol-bis (2-aminoethylether)-N,N,N',N'-tetraacetic acid, 4 mM potassium chloride, 0.3 mM guanosine 5'-triphosphate magnesium salt, and 0.5% biocytin. Recordings were digitized at 200 kHz, Bessel filtered at 10 kHz, and stored using the NeuroData Without Borders format[52]. Before analysis, data were corrected for a measured liquid junction potential of −14 mV.

Morphological reconstruction was performed as described in the Allen Cell Types Database[53]. After recording and fixation, a horseradish peroxidase reaction with diaminobenzidine as a chromogen was performed to visualize the biocytin-filled cells. Cells were imaged on an upright bright-field microscope with a 63× oil-immersion objective lens, and image data were collected as z-stacks. These images underwent automatic 3D neuron reconstruction using the Vaa3D software[54]. Following this, the reconstruction was extensively manually corrected and curated to produce an accurate representation of the somatic and dendritic morphology of the cell that was saved in the SWC format.

**Electrophysiological features.** Electrophysiological features of the recordings were analyzed to provide training targets for the model optimization procedure. The approach of Druckmann et al.[10] was used to evaluate the performance of a model by comparing the model and experimental values of specific features computed from the somatic voltage trace in response to somatic current injection; according to that study, feature comparison was more robust than direct matching of the experimental voltage trace. The models were optimized using responses to long (1–2 s) step current injections, which have been shown to generalize better to other stimulus types than models trained on ramp or noise stimuli[55]. A set of 12 features were computed from experimental voltage traces: the average firing frequency, action potential peak, fast trough depth, slow trough depth, time of slow trough (as a fraction of the ISI), action potential width at half-height (measured from trough to peak), the latency to the first action potential, the duration of the first ISI, the coefficient of variation of the ISIs, the average ISI, and the adaptation index (average difference between consecutive ISIs divided by their sum). The features of individual action potentials were averaged across all action potentials in the response. These features were computed using the Allen SDK software package[56].

**Simulations.** Simulations were run using NEURON 7.4[57]. The SWC file of a target neuron was imported into NEURON with its built-in Import3d function. This function converted the spherical soma representation in the SWC file into a cylinder with equivalent membrane area (as NEURON does not support spherical sections). Since the extent of axonal reconstructions varied considerably across cells, any axonal segments present in the reconstruction were removed, and a synthetic axon initial segment (60 μm length, 1 μm diameter) was attached to the soma.

To estimate the passive properties of the neuron, small, subthreshold responses to 0.5 ms step current injections of ±200 pA were directly fit with an entirely passive model[22, 23] using the PRAXIS method of NEURON's multiple run fitter. This method varied uniform values of the specific membrane capacitance ($C_m$), specific membrane resistance ($R_m$), and specific intracellular resistivity ($R_i$) to best match the experimental responses (see Fig. 2a). Once these values were obtained, $R_i$ was fixed during the genetic algorithm optimization procedure. For aspiny neurons, $C_m$ was also fixed to the value obtained by the direct fit, but for spiny neurons, it was assumed that the dendrites had a higher effective specific membrane capacitance due to the extra membrane area of dendritic spines (which were not explicitly represented). Therefore, the value of $C_m$ from the direct fit was used to calculate a new, higher value of $C_m$ for the dendritic compartments while assuming that the somatic and axonal compartments had a $C_m$ of 1 μF cm$^{-2}$. In addition, when data were not available to perform this fitting procedure for a given cell, standard values were used instead ($R_i = 100$ Ω cm, somatic, axonal, and aspiny $C_m = 1$ μF cm$^{-2}$, spiny dendritic $C_m = 2$ μF cm$^{-2}$). These values represent the effective values for the morphologies used in the simulations, but the actual values

of these parameters in the intact cell are better estimated when the axon is included (Supplementary Fig. 1). Finally, the value obtained for $R_m$ was not used directly, as the values of the conductances (including a passive leak conductance in the dendritic and axonal compartments) were allowed to vary during the later fitting stages.

Active conductances were placed at the soma of the model, while the dendritic compartments remained passive for both computational simplicity and because the somatic whole-cell recordings used here did not provide substantial information about the distribution of active conductances throughout the dendritic tree. In addition, a mechanism describing intracellular Ca$^{2+}$ dynamics was inserted at the soma. This mechanism described the Ca$^{2+}$ dynamics with a first-order ordinary differential equation that modeled the entry of Ca$^{2+}$ via transmembrane currents into a 100 nm submembrane shell with buffering and pumping. The parameters of this mechanism were the time constant of Ca$^{2+}$ removal and the binding ratio of the buffer, and these parameters were allowed to vary during optimization. As mentioned above, a passive leak conductance was also inserted into all sections, and its reversal potential was fixed to the experimentally measured reversal potential of the cell.

Given that the dendrites in these models were passive, for a subset of cells, models with simplified ball-and-stick morphologies were optimized and compared to models with detailed dendritic morphologies (Supplementary Fig. 2). The resulting ball-and-stick models had reduced simulation times but in most cases did not exhibit as good fits to the experimental data.

The set of active conductances were first based on those used by Hay et al.[13], with adjustments in some cases (see Fig. 3). They include multiple voltage-gated sodium, potassium, and calcium conductances; a calcium-activated potassium conductance; and a hyperpolarization-activated cation conductance. The descriptions of these conductances can be found in Supplementary Information; some of the kinetic parameters were adjusted to match more closely the publications upon which the conductance models were based. The maximum densities of the condutances were allowed to vary during optimization, while their kinetic parameters were fixed. All together, 15 or 16 free parameters were adjusted during optimization: 10 active conductance densities, 2 Ca$^{2+}$ dynamics parameters, and 3 or 4 leak conductance densities (depending on the presence of apical dendritic compartments, which had a separate leak conductance density from the other dendritic compartments).

Simulations were run in NEURON using its variable time step method. The equilibrium potentials of Na$^+$ and K$^+$ were based on the internal and external solutions used in the original experiments ($E_{Na} = +53$ mV, $E_K = -107$ mV). The equilibrium potential of Ca$^{2+}$ was calculated at each time step using the Nernst equation with [Ca$^{2+}$]$_o$ = 2 mM and the current value of [Ca$^{2+}$]$_i$ (resting [Ca$^{2+}$]$_i$ = 100 nM). The reversal potential for the $I_h$ conductance was set to −45 mV[29]. The temperature of the simulation was set to 34 °C to match the experiments, and the kinetics of conductance models were scaled with a $Q_{10}$ of 2.3.

**Optimization.** During model fitting, conductance densities were varied to optimize the match of the features of the model to the original experimental data using a genetic algorithm[10, 13]. The genetic algorithm was implemented using the DEAP Python library[21] and the Python interface of NEURON[58]. The procedure iterated toward a set of 15–16 model parameters that matched features of the experimental data within a given tolerance. In the genetic algorithm scheme, an "organism" was a set of parameters, and a population of 1200 organisms was used over 500 generations for each fitting stage. This type of optimization procedure does not guarantee that a global minimum for the fitting error will be found, but the procedure was repeated with five different random seeds, which each produced different results that were compared at the end of fitting. The procedure was run on a local high-performance computing cluster; a single fitting stage typically took between 1 and 4 h using 10 nodes with 24 cores each (240 total cores).

Model fitting was separated into two stages (see Fig. 2). The first stage attempted to match seven electrophysiological features, while the second stage added five more features. After stage 1 was run five times with different seeds, the final population that contained the organism with the lowest error was used as the starting population for each of the stage 2 fits with different seeds. The set of parameters that resulted in the lowest error out of the five stage 2 runs was selected as the representative model for that cell.

The error of a given model for a given feature was calculated as an absolute z-score[10, 13]. The standard deviation for the z-score was calculated from the trial-to-trial variability in a feature when the stimulus was repeated multiple times during the experiment. The absolute z-scores from all the target features were averaged together to provide a single-objective error function for the fit procedure.

In some cases, the experimentally measured variability of a given feature was extremely low, which in effect caused that feature to have a much higher influence on the fit at the expense of the other features. To compensate for that, each feature had a minimum tolerance level that was deemed acceptable (Supplementary Table 1), and when the experimentally measured standard deviations fell below that tolerance, the tolerance value was used to calculate the absolute z-score instead. These tolerance values were also used in the cases when repeated trials of the same stimulus were not available for a given cell.

Depolarization block at relatively low stimulus intensities was sometimes observed in models that had low errors on the training step (see Fig. 4). This outcome was avoided by an additional automated procedure that evaluated a single high-amplitude current step in addition to the training step during optimization

and heavily penalizing organisms with responses that failed to repolarize. After the optimization was complete, models with low error were also checked for failure to repolarize during the pink noise stimulus used during the experiments; models that exhibited depolarization block on that stimulus were discarded, and the model with the next lowest error was selected and evaluated until an acceptable model was identified. Cells that had a best-fitting model with an average training error over 3 (i.e., averaging over three standard deviations away from the mean feature values of the actual cell) were deemed as not well fit, and those models were not included in the model set for this study or publicly released (see "Data availability" section). Models with an average training error between 2 and 3 were manually inspected before inclusion; a small number of these models exhibited clear problems (such as highly distorted spike shapes) and were also excluded from the model set and public release.

**Model generalization and analysis**. Model generalization was assessed by applying additional stimulus protocols that were used in the original experimental recordings. The degree to which the model matched the responses of the original neuron to the pink noise stimuli was assessed by calculating the ratio of the trial-to-trial explained variance of the spike times[39], where the explained variance (EV) between two peristimulus time histograms (PSTHs) is defined as

$$\mathrm{EV}(\mathrm{PSTH}_1, \mathrm{PSTH}_2) = \frac{\mathrm{var}(\mathrm{PSTH}_1) + \mathrm{var}(\mathrm{PSTH}_2) - \mathrm{var}(\mathrm{PSTH}_1 - \mathrm{PSTH}_1)}{\mathrm{var}(\mathrm{PSTH}_1) + \mathrm{var}(\mathrm{PSTH}_2)}.$$
(1)

PSTHs were calculated from individual spike trains by convolving with a Gaussian with a width at a particular time window resolution. The explained variance was calculated over a range of time window resolutions from 0.5 ms to 20 ms. The findings across models were similar at different resolutions, and the EV with time window of 10 ms (as representative at the center of the evaluated range) was used in summary plots (see Fig. 6).

The slope of the *f*–*I* curve was measured by a linear least-squares fit of the average rates of the suprathreshold responses during 1 s current steps.

**Classification**. Supervised classification was performed with a support vector machine with a linear kernel that was evaluated via five-fold cross-validation after recursive feature elimination with cross-validation. Classes were weighted based on their frequencies to compensate for differences in the number of samples available. The classifier was trained to predict either general cell class (either pyramidal cells or interneurons from one of the major transgenically labeled classes) or a specific electrophysiological class identified by an unsupervised clustering procedure performed on electrophysiological features taken from a broad set of cells in the Allen Cell Types Database[39], which found 11 different clusters. That data set, performed on cells in which generalized leaky integrate-and-fire neuron models were fit, partially overlapped with the cells modeled in this study, and only the 10 clusters (out of 16 total) that contained at least 5 cells from this study were used in the classification analysis here.

Morphology-related parameters of the models were calculated for use in classification. These included the total capacitance of the model, the ratio of the somatic capacitance to the capacitance of the dendritic compartments, and the average attenuation from the soma to the dendritic tips. The latter was calculated by voltage-clamping the soma of the model to 10 mV above its resting potential, then calculating and averaging together the voltage change measured at each dendritic tip.

**Code availability**. The code for electrophysiological feature analysis and model simulation is available as part of the open-source Allen SDK repository (alleninstitute.github.io/AllenSDK). Code for model optimization is also available as an open-source repository (github.com/alleninstitute/biophys_optimize).

**Data availability**. The electrophysiological and morphological data supporting the findings of this study and the models generated for this study are available in the Allen Cell Types Database, celltypes.brain-map.org. A subset of the models is also available via the ModelDB repository[48], senselab.med.yale.edu/ModelDB. Morphological data are also available through the NeuroMorpho.org repository[47], neuromorpho.org.

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

## Acknowledgements

We thank E. Hay and I. Segev for advice regarding the optimization of biophysically detailed models. We thank M. Hines for assistance with the NEURON software. We wish to thank the Allen Institute founders, Paul G. Allen and Jody Allen, for their vision, encouragement and support. This work was funded by the Allen Institute for Brain Science.

## Author contributions

N.W.G. and A.A. designed the study. N.W.G. developed and optimized the models, performed data analysis, and prepared the manuscript. J.B. and S.A.S. collected and analyzed experimental data. D.F. wrote analysis, optimization, and simulation code with N.W.G. H.Z. conceived and managed the electrophysiological data production pipeline. M.H. directed the primary group conducting the study. A.A. supervised the study and edited the manuscript. C.K. conceived of the project, gave conceptual advice, and edited the manuscript.

## Additional information

**Competing interests:** The authors declare no competing financial interests.

