## [Peer Review File · Nature Communications]

Reviewers' comments:

Reviewer #1 (Remarks to the Author):

Overall: This is a very strong paper with novel methods that will be widely useful for neuroscientists. This work and tools offered are highly beneficial to the research community because it is currently very difficult, time-consuming, and error-prone to systematically and objectively develop detailed computer models of different classes of neurons. The methods proposed here address several problems with previous attempts to solve the model development issue and do so in a streamlined way. The authors have also used their large sets of models and data to explore whether and how specific channels contribute to different dynamical features - an important and creative result in its own right. I only have minor concerns mentioned below, and provide suggestions for possible additional, related studies and discussion points.

Minor concerns:

figure 1 -

b - does this code automatically run in parallel?

c - it's not clear what the little 'dots' under the voltage traces are
please clarify both of these issues in the figure legend

lines 125 - 131 -- can the authors clarify why combining features using a z-score is a 'better' approach than using a more traditional multiobjective evaluation with pareto archival or how their approach may compare to that taken by Rumbell et al 2016 (Automatic fitness function selection for compartment model optimization). This could be mentioned in the discussion.

lines 132- 136 -- what is the threshold used to determine whether a model is good enough to enter the final database of models?

lines 143 -- 150 -- are the passive parameters uniformly set in all sections of the neuronal models? if yes/no, why? This is touched upon in methods but may be useful here too.

line 177 -- why are active conductances only placed at the soma? why not estimate conductances at dendrites too based on known (from literature, other experiments) distribution of channels in dendrites?

lines 198 -- 200 -- the staged fitting approach is interesting - can the authors comment on why specific features were used for stage 1 vs 2 - does it have to do with a coarse to fine fitting procedure?

paragraph starting with line 207 -- interesting approach and sounds reasonable - is it also possible that the location of spike initiation influences the shape of the action potential? for example, if spikes were generated in a modeled axon, might AP shape more accurately replicate the experimental data?

The SVM classification approach is interesting but I tend to agree with the authors that the morphology is likely to play an important role in distinguishing different model classes - could the authors comment on whether utilizing statistical descriptions of morphological features are likely to improve these classifications? perhaps a classification using such features could be attempted?

Reviewer #2 (Remarks to the Author):

Systematic, high-throughput generation of biophysically detailed models for diverse cortical neuron types

Gouwens et al

In this study the authors develop a useful resource of models of 170 cells from experimental data from the Allen Cell Types Database. This will be useful for many groups doing detailed modeling, and also is intended for network models. The authors show that their approach to using the panel of electrophysiological readouts from the Allen Brain Project yields models that fire similarly to their experimental counterparts, and are classified in much the same way. Many papers have optimized individual models, but this study draws upon a much larger dataset and one having systematic measurements of cellular properties.

Major comments.

1. The authors make an important simplification by placing all active conductances in the soma (Methods section, line 531 onwards). The rationale is that the somatic recordings don't provide much information about dendritic conductances. Nevertheless, it may be less of an assumption and approximation to take published dendritic channel distributions in this model. The point being that even imperfect dendritic channel distributions are unlikely to be as wrong as passive dendrites. It should be possible for the authors to systematically evaluate this possibility by redoing the optimization, but now with a publication-based distribution of ion channels in the dendrites. The optimization could happen as before, with just the somatic ion channels being varied. The authors refer to an SfN abstract but in the current study it would be worth at least estimating how much the optimized somatic channel densities differ between models with and without active dendrites. This shouldn't take too long for a couple of test cases, by the authors own account of simulation times.

2. Given that the model ignores dendritic channels, could the same spiking properties be obtained with a ball-and-stick model with just the active soma as before? The dendrite could be a single equivalent cylinder with an electrotonic size (number of length constants) equal to that of the morphologically detailed model. This would hugely reduce the computational cost of the model.

3. Classifier accuracy from model parameters: Page 17. It is interesting that this performs worse than the model itself, and I wonder if the classification would be much better if the authors included a couple of key morphologically-related parameters in the classification: the electrical size of the cell (measured in number of electrotonic length constants), and the ratio of somatic to dendritic capacitance.

4. It would be nice if the authors could provide the models in NeuroML format.

Minor comments

Typo: line 338

Reviewer #3 (Remarks to the Author):

The manuscript "Systematic, high-throughput generation of biophysically detailed models for diverse cortical neuron types" describes modeling study base on the Allen Cell Types Database in which 170 multi-compartment Hodgkin-Huxley models for different cells/cell types are generated (active conductances in the soma, passive dendrites). The authors employ a genetic algorithm for parameter optimization mostly of conductance values of otherwise fixed channel kinetics. The study closes on analyzing whether the diversity seen in the original experimental data is preserved in the generated models.

The presented manuscript reflects a systematic and thorough work of a well known group of scientists. It leverages the experimental resources put online for public use by the Allen Institute to generate a sizable set of models with few prior assumptions for a diverse set of cells. It makes models and software used to prepare the models publicly available. At the same time, the study adds little beyond the state of the art on how this is done: it uses established methods to generate the models (long step-current stimuli, feature-based + z-score error function, genetic algorithm, search space reduction to mostly adjusting conductance values) and shows classical generalization (e.g. ramp stimulus, nosy stimuli). It uses previously assembled set of conductances that were shown effective for modeling a range of different cell types and extends/augments this marginally to create firing types that show slightly faster time constants.

Creating new models of a diverse set of neurons, sharing them with the community and making the code available is a very good thing. Especially, if it is done in a systematic, reproducible manner. But there are several aspects that make me conclude that this study is not material for Nature Communication but rather an excellent addition to the body of literature on single cell modeling in a speciality journal. Here is why:

- Related efforts: While the study makes some mention of other papers that have created sizable amounts of detailed neuron models, it is particularly surprising that another effort from scientists of the same organization, which has published a substantial albeit smaller set of 40 models on the same or similar experimental data, goes unmentioned (maybe because it is only a technical white paper?). While of course the reference to this technical white paper can be fixed, this observation takes away from the novelty of this present study. This is not to say that the results of the study are not worth to be published but it points into the direction that we have a manuscript for a speciality journal in hand rather than Nature communication. It also raises the question whether a second study on building models on the same dataset shouldn't compare itself to the results of the previous study.
- Community embedding: While the paper is systematic, makes data/models/tools available and gives a good account of related literature, it essentially shows little recognition of well established community resources such as Neuromorpho.org (for morphologies) or ModelDB (for models) and essentially reimplements previously published open source tools (see e.g. Friedrich et al., Van Geit et al. as already cited in the study). While the former part of this comment could probably be addressed by uploading/linking morphologies also in neuromorpho and models in ModelDB, it is of course unreasonable to ask the authors to drop their tool and switch to another. But that said, it is clear that this study is to be seen as contribution to a specific community and should be judged/embedded in the respective speciality journal. Highlighting the (bottom-line) reimplementations and application of previously published methods in Nature Communication sounds inappropriate.

Furthermore, to the study design itself I have the following remarks:

- Single-objective vs. multi-objective: the authors report that they have tested a multi-objective fitness function but that it converged slower than the single-objective approach. While this may be a reasonable conclusion, I believe that the study inherits some subsequent problems from this decision that would be interesting to be looked at closer (e.g. by a more quantitative comparison of a single objective run vs a multi-objective run). The particular points are 1) as the authors report, without any special precaution, a sizable portion of their models exhibit a block, ie. they get stuck at a specific membrane depolarization. The authors remedy that fact by detecting and heavily penalizing these models. My suspicion would be that this is at least to some degree a consequence from not giving the algorithm enough power to identify that such a trace simply is not matching experimental targets (the averaging of different features may downplay the power of a feature such as average firing frequency, ISI that in principle should penalize such behavior). So, it would be good to know whether this special hack of the penalization of blocked neurons can be avoided by a better fitness function? 2) the authors observe that some features are more easily fitted than others (figure 5c). The authors leave it open whether this is a problem of the search space or the relative weight of the features. Once more, multi-objective optimization should be able to help clarify. The omission of multi-objective optimization on the sole ground of performance is therefore not fully convincing and should be revisited.
- Comparison to prior results: as mentioned before, scientists from the same institute have also generated a set of 40 detailed models on the same data set. It would be important for this study to analyze in more detail how the presented models score in relation to those other models.
- Models and Cell type characterization (Figure 7): this part of the paper I find the weakest. I can understand why the authors wanted to add this section, but I think the results are also not really convincing and possibly they are using the wrong tool to do so. But what are my fundamental issues here: The authors want to show diversity. For that they train an SVN to predict this (externally defined) diversity from the features they are extracting from the experimental trace, which they succeed to a very limited level. In 7a1 they claim 79% accuracy (which is not stellar), which seems to be mostly based on the success in pyramidal cells, but for most interneurons is much worse than that. Then they try the same on the features extracted from the model (7a2), with quantitatively similar results. Their claim is that this means that the models maintain distinctiveness. However, I am not sure this argument holds. Training an SVN to predict something can pick up on many aspects of input data (e.g. the predictive property could be that a particular class can be predicted most easily from a feature being 3 sigma away). So, without further analysis it is not clear that those findings are actually comparable (unless they don't retrain?). Then, when doing this one more time from model parameters (7a3), the absolute result is bad (64%) and it is not clear to me why it should work in the first place as solutions for modeling biophysical models are known to typically be non-unique. So, whatever is the result, it could purely be incidental. When doing this on a probably better defined set of classes (7b), the results are worse and the above systematic criticism holds. To give this part full credit, it might be furthermore necessary to have access to the methods described in their reference 36 (where it says submitted). I think overall, the argument of the models recreating diversity would be better served by a better quantitative analysis of how the models score in their features with respect to the experimental traces.

In conclusion, I can very much see this manuscript published (after addressing above comments) in a speciality journal, but not in its current form in Nature Communications.

Some minor comments:

- figure 1: is illegible (too small) and should be reworked
- figure 2: panels e,f - reference should be made in the caption to how the error is defined
- legend fig7 & line 545: I think something goes wrong with the citations here: in the legend it says

“clustering from 36” and in line 545 “those used by 13” probably should read “clustering from [36]/those used by [13]” or similar depending on the citation style

Reviewer #4 (Remarks to the Author):

The authors fit biophysical neuron models to the Allen Institute cell types data of 170 neurons (available at celltypes.brain-map.org/), with the aim of providing an automated pipeline for high-throughput fitting of biophysical neuronal models. There are interesting results in this work (namely the fitting of biophysical models to the high throughput data), but similarly to the companion paper on generalized leaky integrate-and-fire neurons, it seems hard to discern the key contribution of the study. As the aim here is not to describe the Allen cell-type data per se, or to offer a fundamentally new approach for fitting biophysical neuron models (which has been the subject of several previous studies, e.g. by Druckmann et al.), the question I faced was whether the application of a new and “modern” pipeline to high-throughput data of neuronal cell types together represent an important novel contribution.

The title of the study is informative in this regard: specifically, the stated focus is on “high-throughput generation of biophysically detailed models for diverse cortical neuron types”, but the presented work is probably more accurately described as “generation of biophysically detailed models for high-throughput diverse cortical neuron types”. In other words, the data acquisition is high-throughput not the modeling. The distinction seems important for the main contribution of the paper. Specifically, if the authors wish to claim that the fitting is high-throughput, which would be an important achievement, they should back up such claims with more quantitative results, specifically benchmarking over previous fitting approaches.

Admittedly, the fitting of biophysical neuronal models is highly nontrivial. However, even taking this into account, the multiple local optima associated with each model make the interpretation hard. The conceptual advantage of biophysical models is in the ease of interpretability of the model parameters; but this advantage disappears when there are distinct local optima. As the authors also describe, the specific parameter values do not necessarily reproduce other properties of neurons, and require a manual and somewhat ad hoc check and penalization of models which do not exhibit “best match” to additional neuronal dynamics. This step goes against the very idea of automated and high-throughput fitting. Furthermore, as the authors do not consider all potentially interesting properties of neuronal activity (as described in e.g. Izhikevich’s 2004 paper), and so a natural question arises whether the “best match” models may exhibit artificial dynamics for other untested neuronal features – potentially resulting in the need for more ad hoc penalizations.

If the authors wish to achieve the laudable goal of an automated and high-throughput fitting of biophysical neurons, they should consider how to improve their current pipeline, such that there is a clear and well interpreted global optimum, or that the set of local optima are close to each other in parameter space, and ad hoc “best match” checks become unnecessary. The utilized genetic algorithm is quite simple, and I wonder if the authors could consider a more sophisticated algorithm for this hard problem. This is not my area of expertise but I suspect that, given the importance of nonlinear multiobjective optimization in computational science more generally, there are bound to exist more sophisticated yet more effective algorithms; perhaps it is worth to consult an optimization expert (not necessarily in neuroscience) to try and produce more robust fits without distinct local optima. This is presently the key roadblock to the authors aim of achieving high-throughput and automated fitting.

Other: The clarity of Figure 1 could be substantially improved. Specifically, I suggest to make it more

explicit in Figure 1a which features are fitted, and which generalizations are tested, to clarify that the test and validation are independent. Panels b and c are uninformative, and could be moved to supplementary information (I suggest to include a larger chunk of self-contained code in SI as a standalone text, rather than as part of a figure). More generally, the size of legends in all figures is too small and should be increased.

Response to Reviewer 1:

Overall: This is a very strong paper with novel methods that will be widely useful for neuroscientists. This work and tools offered are highly beneficial to the research community because it is currently very difficult, time-consuming, and error-prone to systematically and objectively develop detailed computer models of different classes of neurons. The methods proposed here address several problems with previous attempts to solve the model development issue and do so in a streamlined way. The authors have also used their large sets of models and data to explore whether and how specific channels contribute to different dynamical features - an important and creative result in its own right. I only have minor concerns mentioned below, and provide suggestions for possible additional, related studies and discussion points.

Minor concerns:

figure 1 -

b - does this code automatically run in parallel?

*c - it's not clear what the little 'dots' under the voltage traces are
please clarify both of these issues in the figure legend*

We have reworked Figure 1 and its figure legend to indicate that the optimization module of our code runs in parallel automatically, and to provide annotations of the screenshot in Fig. 1c to clarify that the “dots” indicate the spike times of the traces above.

lines 125 - 131 -- can the authors clarify why combining features using a z-score is a 'better' approach than using a more traditional multiobjective evaluation with pareto archival or how their approach may compare to that taken by Rumbell et al 2016 (Automatic fitness function selection for compartment model optimization). This could be mentioned in the discussion.

lines 132- 136 -- what is the threshold used to determine whether a model is good enough to enter the final database of models?

lines 143 -- 150 -- are the passive parameters uniformly set in all sections of the neuronal models? if yes/no, why? This is touched upon in methods but may be useful here too.

We have revised the manuscript text to address the use of single-objective fitting in the Discussion section, to specify the model inclusion criteria, and to clarify how we are setting passive parameters in the Results section.

line 177 -- why are active conductances only placed at the soma? why not estimate conductances at dendrites too based on known (from literature, other experiments) distribution of channels in dendrites?

We have added an additional reference to a white paper describing a set of models with active conductances in the dendrites built from data in the Allen Cell Types Database. Examining the differences between our model set and this one with active dendritic conductances will be of high interest. However, a comparison that appropriately considers different aspects of model performance for different configurations of active dendritic conductances and takes into account the methodological differences in model optimization requires much additional work warranting a separate, future study.

[Redacted]

lines 198 -- 200 -- the staged fitting approach is interesting - can the authors comment on why specific features were used for stage 1 vs 2 - does it have to do with a coarse to fine fitting procedure?

Yes, this staged approach does begin with coarser features and then adds finer features. We have revised the text to clarify our staged fitting approach.

paragraph starting with line 207 -- interesting approach and sounds reasonable - is it also possible that the location of spike initiation influences the shape of the action potential? for example, if spikes were generated in a modeled axon, might AP shape more accurately replicate the experimental data?

We agree that the location of spike initiation will affect spike shape, and we address that point in the Discussion. For example, the sharp rise of the spikes seen in the experimental data but not the model is likely due to the initiation of the spike in the axon.

The SVM classification approach is interesting but I tend to agree with the authors that the morphology is likely to play an important role in distinguishing different model classes - could the authors comment on whether utilizing statistical descriptions of morphological features are likely to improve these classifications? perhaps a classification using such features could be attempted?

We appreciate this helpful suggestion to attempt classification of cells after including morphological parameters, and we have revised Fig. 7 to illustrate this additional classification. As described in the revised text, we calculated the total capacitance, the somatic-to-dendritic capacitance ratio, and the average attenuation from soma to dendritic tip for each model. We found that adding these morphology-related

parameters does indeed improve classification performance over model parameters alone. Also, Fig. 7b now uses updated clustering results from Teeter et al. (submitted).

Response to Reviewer 2:

In this study the authors develop a useful resource of models of 170 cells from experimental data from the Allen Cell Types Database. This will be useful for many groups doing detailed modeling, and also is intended for network models. The authors show that their approach to using the panel of electrophysiological readouts from the Allen Brain Project yields models that fire similarly to their experimental counterparts, and are classified in much the same way. Many papers have optimized individual models, but this study draws upon a much larger dataset and one having systematic measurements of cellular properties.

Major comments.

1. The authors make an important simplification by placing all active conductances in the soma (Methods section, line 531 onwards). The rationale is that the somatic recordings don't provide much information about dendritic conductances. Nevertheless, it may be less of an assumption and approximation to take published dendritic channel distributions in this model. The point being that even imperfect dendritic channel distributions are unlikely to be as wrong as passive dendrites. It should be possible for the authors to systematically evaluate this possibility by redoing the optimization, but now with a publication-based distribution of ion channels in the dendrites. The optimization could happen as before, with just the somatic ion channels being varied. The authors refer to an SfN abstract but in the current study it would be worth at least estimating how much the optimized somatic channel densities differ between models with and without active dendrites. This shouldn't take too long for a couple of test cases, by the authors own account of simulation times.

In response to the reviewer's comments about models with active conductances in the dendrites, we have added an additional reference to a white paper describing a set of models with active conductances in the dendrites built from data in the Allen Cell Types Database. We agree that examining the differences between these model sets will be of high interest. However, a comparison that appropriately considers different aspects of model performance for different configurations of active dendritic conductances and takes into account the methodological differences in model optimization requires much additional work warranting a separate, future study. Indeed, we have begun work on such a study, and our preliminary comparisons between the passive-dendrite models presented here and the active-dendrite models in the Allen Cell Types Database do **not** show significant improvements by the latter in reproducing features of the somatic voltage responses.

2. Given that the model ignores dendritic channels, could the same spiking properties be obtained with a ball-and-stick model with just the active soma

as before? The dendrite could be a single equivalent cylinder with an electrotonic size (number of length constants) equal to that of the morphologically detailed model. This would hugely reduce the computational cost of the model.

We thank the reviewer for this excellent suggestion, and we have now optimized ball-and-stick alternative models for a subset of cells and compared them to the models with detailed morphologies. The results are shown in Supplementary Fig. 2. We find that with the same optimization procedure, in most cases the ball-and-stick models do not produce as good of fits as the models with detailed morphologies (Supplementary Fig. 2a). We also find that the simulation times are ~50% shorter with the ball-and-stick models (Supplementary Fig. 2b), so the computational savings are relatively modest. It is possible that future work could improve the ball-and-stick optimization methods to produce simplified models with errors closer to those of the detailed morphology models.

3. Classifier accuracy from model parameters: Page 17. It is interesting that this performs worse than the model itself, and I wonder if the classification would be much better if the authors included a couple of key morphologically-related parameters in the classification: the electrical size of the cell (measured in number of electrotonic length constants), and the ratio of somatic to dendritic capacitance.

We appreciate this helpful suggestion to attempt classification of cells after including morphological parameters, and we have revised Fig. 7 to illustrate this additional classification. As described in the revised text, we calculated the total capacitance, the somatic-to-dendritic capacitance ratio, and the average attenuation from soma to dendritic tip for each model. We found that adding these morphology-related parameters does indeed improve classification performance over model parameters alone. Also, Fig. 7b now uses updated clustering results from Teeter et al. (submitted). Thank you for the suggestion of this approach.

4. It would be nice if the authors could provide the models in NeuroML format.

We are aware that code has been written (independently of this study) to convert our models to the NeuroML format (<https://github.com/OpenSourceBrain/AllenInstituteNeuroML>). We are also currently working on enabling support for the NeuroML format for our modeling projects. Therefore, we will explore the possibility of distributing our single-cell models in this format alongside the currently available files.

Minor comments

Typo: line 338

This has been corrected.

Response to Reviewer 3:

The manuscript “Systematic, high-throughput generation of biophysically detailed models for diverse cortical neuron types” describes modeling study base on the Allen Cell Types Database in which 170 multi-compartment Hodgkin-Huxley models for different cells/cell types are generated (active conductances in the soma, passive dendrites). The authors employ a genetic algorithm for parameter optimization mostly of conductance values of otherwise fixed channel kinetics. The study closes on analyzing whether the diversity seen in the original experimental data is preserved in the generated models.

The presented manuscript reflects a systematic and thorough work of a well known group of scientists. It leverages the experimental resources put online for public use by the Allen Institute to generate a sizable set of models with few prior assumptions for a diverse set of cells. It makes models and software used to prepare the models publicly available. At the same time, the study adds little beyond the state of the art on how this is done: it uses established methods to generate the models (long step-current stimuli, feature-based + z-score error function, genetic algorithm, search space reduction to mostly adjusting conductance values) and shows classical generalization (e.g. ramp stimulus, noisy stimuli). It uses previously assembled set of conductances that were shown effective for modeling a range of different cell types and extends/augments this marginally to create firing types that show slightly faster time constants.

Creating new models of a diverse set of neurons, sharing them with the community and making the code available is a very good thing. Especially, if it is done in a systematic, reproducible manner. But there are several aspects that make me conclude that this study is not material for Nature Communication but rather an excellent addition to the body of literature on single cell modeling in a speciality journal. Here is why:

- Related efforts: While the study makes some mention of other papers that have created sizable amounts of detailed neuron models, it is particularly surprising that another effort from scientists of the same organization, which has published a substantial albeit smaller set of 40 models on the same or similar experimental data, goes unmentioned (maybe because it is only a technical white paper?). While of course the reference to this technical white paper can be fixed, this observation takes away from the novelty of this present study. This is not to say that the results of the study are not worth to be published but it points into the direction that we have a manuscript for a speciality journal in hand rather than Nature communication. It also raises the question whether a second study on building models on the same dataset shouldn't compare itself to the results of the previous study.

We thank the reviewer for these comments. First, we should point out that the technical white papers that the referee is referring to are meant (associated with the Allen Cell Types Database) for both the active-dendrite biophysical models and the passive-dendrite models – the latter being the subject of our present manuscript. These technical white papers are not formal peer-reviewed publications and are provided to guide the users of all of our data sets, including the Allen Cell Types Database. The data and associated models presented in these white papers are released publicly prior to any formal publication, in accordance with the long-standing commitment of the Allen Institute to open science. For both types of models, the white papers contain basic information on methodology, but little description of their generalization, characterization, or related scientific questions as none of that extensive work had taken place at the time of our first model release. Now that we have done this, we wrote a more comprehensive manuscript, suitable for peer review and intended to be a resource for more sophisticated users of our data or scientists who would like to use automatic methods for optimizing neuronal models similar to what we have done. The researchers working on the active-dendrite models are proceeding similarly.

We have added a reference to the white paper about the active-dendrite models, alongside the reference we included in our original submission to a conference abstract on the same topic by the same group of researchers.

The reviewer also asked about a comparison of our models to the active dendrite models built from the same data set. We agree that examining the differences between these model sets will be of high interest. However, a comparison that appropriately considers different aspects of model performance for different configurations of active dendritic conductances and takes into account the methodological differences in model optimization requires much additional work warranting a separate, future study.

[Redacted]

- Community embedding: While the paper is systematic, makes data/models/tools available and gives a good account of related literature, it essentially shows little recognition of well established community resources such as Neuromorpho.org (for morphologies) or ModelDB (for models) and essentially reimplements previously published open source tools (see e.g. Friedrich et al., Van Geit et al. as already cited in the study). While the former part of this comment could probably be addressed by uploading/linking morphologies also in neuromorpho and models in ModelDB, it is of course unreasonable to ask the authors to drop their tool and switch to another. But that said, it is clear that this study is to be seen as contribution to a specific community and should be judged/embedded in the respective speciality journal. Highlighting the (bottom-line) reimplementation and application of previously published methods in Nature Communication sounds inappropriate.

We appreciate the comments from the reviewer regarding data sharing and community standards/resources. The reviewer correctly points out that it is useful to mention other widely-used community data-sharing resources; we have corrected this by now referencing them in the text and explaining that our data and models are also available from both ModelDB and NeuroMorpho in the Data Availability section in Methods. We note that the models in ModelDB are a subset of those in the study based on the first public release of the Allen Cell Types Database, though additional models could be added with the same methods that the ModelDB maintainers developed for that first ingestion.

Furthermore, to the study design itself I have the following remarks:

- Single-objective vs. multi-objective: the authors report that they have tested a multi-objective fitness function but that it converged slower than the single-objective approach. While this may be a reasonable conclusion, I believe that the study inherits some subsequent problems from this decision that would be interesting to be looked at closer (e.g. by a more quantitative comparison of a single objective run vs a multi-objective run). The particular points are 1) as the authors report, without any special precaution, a sizable portion of their models exhibit a block, ie. they get stuck at a specific membrane depolarization. The authors remedy that fact by detecting and heavily penalizing these models. My suspicion would be that this is at least to some degree a consequence from not giving the algorithm enough power to identify that such a trace simply is not matching experimental targets (the averaging of different features may downplay the power of a feature such as average firing frequency, ISI that in principle should penalize such behavior). So, it would be good to know whether this special hack of the penalization of blocked neurons can be avoided by a better fitness function? 2) the authors observe that some features are more easily fitted than others (figure 5c). The authors leave it open whether this is a problem of the search space or the relative weight of the features. Once more, multi-objective optimization should be able to help clarify. The omission of multi-objective optimization on the sole ground of performance is therefore not fully convincing and should be revisited.

In response to the reviewer's comments about multi-objective methods and the check for depolarization block that we describe in our methods, we would like to clarify that the depolarization block check is unrelated to questions of single-objective vs multi-objective methods. The depolarization block was never observed in the responses to the training current step - as mentioned by the reviewer, the firing frequency and ISI features do indeed penalize that behavior (see Fig. 4a, red, left). The depolarization block was only observed when model generalization was assessed on a higher-amplitude test current step (Fig. 4a, red, right). Consequently, the training features used in optimization would carry no information about the presence or absence of the block on the higher-amplitude step, regardless of whether they are combined into a single

objective or evaluated by a multi-objective method. We have also added a section to the Discussion regarding the use of single- vs multi-objective methods.

- *Comparison to prior results: as mentioned before, scientists from the same institute have also generated a set of 40 detailed models on the same data set. It would be important for this study to analyze in more detail how the presented models score in relation to those other models.*

- *Models and Cell type characterization (Figure 7): this part of the paper I find the weakest. I can understand why the authors wanted to add this section, but I think the results are also not really convincing and possibly they are using the wrong tool to do so. But what are my fundamental issues here: The authors want to show diversity. For that they train an SVN to predict this (externally defined) diversity from the features they are extracting from the experimental trace, which they succeed to a very limited level. In 7a1 they claim 79% accuracy (which is not stellar), which seems to be mostly based on the success in pyramidal cells, but for most interneurons is much worse than that. Then they try the same on the features extracted from the model (7a2), with quantitatively similar results. Their claim is that this means that the models maintain distinctiveness. However, I am not sure this argument holds. Training an SVN to predict something can pick up on many aspects of input data (e.g. the predictive property could be that a particular class can be predicted most easily from a feature being 3 sigma away). So, without further analysis it is not clear that those findings are actually comparable (unless they don't retrain?). Then, when doing this one more time from model parameters (7a3), the absolute result is bad (64%) and it is not clear to me why it should work in the first place as solutions for modeling biophysical models are known to typically be non-unique. So, whatever is the result, it could purely be incidental. When doing this on a probably better defined set of classes (7b), the results are worse and the above systematic criticism holds. To give this part full credit, it might be furthermore necessary to have access to the methods described in their reference 36 (where it says submitted). I think overall, the argument of the models recreating diversity would be better served by a better quantitative analysis of how the models score in their features with respect to the experimental traces.*

The reviewer notes that an SVM prediction could be picking up some aberrant aspect of the data to make its predictions and therefore may not demonstrate the points we state in describing Fig. 7. We agree with the reviewer that this is a risk. However, the preceding Figs. 5 and 6 illustrate that the models perform well and do not appear to have extreme behavior that the SVM would potentially use as a basis for prediction. Under these circumstances, we believe the use of SVM in this way is valid.

In conclusion, I can very much see this manuscript published (after addressing above comments) in a speciality journal, but not in its current form in Nature Communications.

Some minor comments:

- figure 1: is illegible (too small) and should be reworked*
- figure 2: panels e,f - reference should be made in the caption to how the error is defined*
- legend fig7 & line 545: I think something goes wrong with the citations here: in the legend it says "clustering from 36" and in line 545 "those used by 13" probably should read "clustering from [36]/those used by [13]" or similar depending on the citation style*

We have reworked Figure 1, added the definitions of the error to Figure 2, and corrected the citation references in the places mentioned.

Response to Reviewer 4:

The authors fit biophysical neuron models to the Allen Institute cell types data of 170 neurons (available at celltypes.brain-map.org/), with the aim of providing an automated pipeline for high-throughput fitting of biophysical neuronal models. There are interesting results in this work (namely the fitting of biophysical models to the high throughput data), but similarly to the companion paper on generalized leaky integrate-and-fire neurons, it seems hard to discern the key contribution of the study. As the aim here is not to describe the Allen cell-type data per se, or to offer a fundamentally new approach for fitting biophysical neuron models (which has been the subject of several previous studies, e.g. by Druckmann et al.), the question I faced was whether the application of a new and “modern” pipeline to high-throughput data of neuronal cell types together represent an important novel contribution.

The title of the study is informative in this regard: specifically, the stated focus is on “high-throughput generation of biophysically detailed models for diverse cortical neuron types”, but the presented work is probably more accurately described as “generation of biophysically detailed models for high-throughput diverse cortical neuron types”. In other words, the data acquisition is high-throughput not the modeling. The distinction seems important for the main contribution of the paper. Specifically, if the authors wish to claim that the fitting is high-throughput, which would be an important achievement, they should back up such claims with more quantitative results, specifically benchmarking over previous fitting approaches.

We appreciate the reviewer’s comments on this topic and understand the concern. It is perhaps fair to say that “high-throughput” implies the ability to produce certain results at scale, which we have demonstrated by generating 170 models of individual cells. Still, this process is necessarily computationally expensive for the biophysical models (as described in the Methods section). To avoid potential misunderstanding, we removed “high-throughput” from the title of our manuscript (which now is “Systematic generation of biophysically detailed models for diverse cortical neuron types”). Indeed, the same systematic procedure was applied to fitting every cell, and all aspects of the method are automated (no manual interventions are required). This point may not have been completely clear in our original text, and we have emphasized it now throughout the manuscript. In addition, the term “high-throughput” is only used in reference to the experimental data pipeline in the text.

Admittedly, the fitting of biophysical neuronal models is highly nontrivial. However, even taking this into account, the multiple local optima associated with each model make the interpretation hard. The conceptual advantage of biophysical models is in the ease of interpretability of the model parameters; but this advantage disappears when there are distinct local optima. As the authors also describe, the specific parameter values do not necessarily reproduce other properties of neurons, and

require a manual and somewhat ad hoc check and penalization of models which do not exhibit “best match” to additional neuronal dynamics. This step goes against the very idea of automated and high-throughput fitting. Furthermore, as the authors do not consider all potentially interesting properties of neuronal activity (as described in e.g. Izhikevich's 2004 paper), and so a natural question arises whether the “best match” models may exhibit artificial dynamics for other untested neuronal features – potentially resulting in the need for more ad hoc penalizations.

We thank the reviewer for the comments and questions about the depolarization block check, and we have revised the text in the Results and Methods sections to improve the clarity of those sections. We now explain in the text more clearly that the depolarization block check is actually an entirely automated procedure and does not require any manual intervention. We also note in the Discussion section the caveat that we only assess the models against the stimuli used in the original experiments, and, thus, as pointed out by the reviewer, the models may not reproduce more complex spiking behavior that the original cells would have exhibited if probed with other protocols.

If the authors wish to achieve the laudable goal of an automated and high-throughput fitting of biophysical neurons, they should consider how to improve their current pipeline, such that there is a clear and well interpreted global optimum, or that the set of local optima are close to each other in parameter space, and ad hoc “best match” checks become unnecessary. The utilized genetic algorithm is quite simple, and I wonder if the authors could consider a more sophisticated algorithm for this hard problem. This is not my area of expertise but I suspect that, given the importance of nonlinear multiobjective optimization in computational science more generally, there are bound to exist more sophisticated yet more effective algorithms; perhaps it is worth to consult an optimization expert (not necessarily in neuroscience) to try and produce more robust fits without distinct local optima. This is presently the key roadblock to the authors aim of achieving high-throughput and automated fitting.

Optimization of biophysical model parameters is a difficult problem, but previous work has found that evolutionary/genetic algorithm variants perform well at this task (e.g., Druckmann et al., 2007; Van Geit et al., 2016; Neymotin et al., 2017), despite no guarantee of finding the global optimum in parameter space. In the early stages of our study, we evaluated multiple optimization techniques, including several genetic algorithm variants as well as other methods, and selected the one that gave the best results. We also empirically observed that different random seeds typically converged to similar locations in the parameter space. Finally, the use of the DEAP toolbox in our code makes it straightforward to implement new optimization methods as they become available, allowing easy testing in pursuit of improved performance.

Thus, it is fair to say that researchers in the field (including ourselves) have tried a considerable variety of optimization approaches; nevertheless, the problem of potential

multiple solutions remains hard. We expect that progress on this issue will come not necessarily from better optimization techniques, but rather better constraints derived from more comprehensive experimental characterization of neurons. Specifically, the state of the art may improve dramatically if simultaneous experimental recordings from multiple locations in a neuron beyond just the soma become more accessible (currently, this is very challenging) and/or if additional information, such as transcriptomic characterization of the recorded neuron, becomes more commonplace.

Other: The clarity of Figure 1 could be substantially improved. Specifically, I suggest to make it more explicit in Figure 1a which features are fitted, and which generalizations are tested, to clarify that the test and validation are independent. Panels b and c are uninformative, and could be moved to supplementary information (I suggest to include a larger chunk of self-contained code in SI as a standalone text, rather than as part of a figure). More generally, the size of legends in all figures is too small and should be increased.

We have added a section to the Discussion regarding the use of single- vs multi-objective optimization methods. We have substantially reworked Figure 1 to improve its legibility and clarity. Along similar lines, we have increased font sizes across all the figures in response to the reviewer's comments.

Reviewers' comments:

Reviewer #1 (Remarks to the Author):

Thank you for addressing my concerns in the previous review. The manuscript is now significantly improved and I believe, warrants publication in Nature Communications.

Reviewer #2 (Remarks to the Author):

Systematic generation of biophysically detailed models for diverse cortical neuron types

Gouwens et al

This revision addresses all my concerns from the original MS.

Minor point:

Figure 6 legend, last 2 lines: Do you mean examples g and h?

Reviewer #3 (Remarks to the Author):

The authors present a revised version of their manuscript, now entitled "Systematic generation of biophysically detailed models for diverse cortical neuron types". They have addressed multiple of the different reviewers' comments and overall substantially improved the manuscript.

Overall, the manuscript now describes much clearer what are the contributions of the study, in particular that they are fitting models for individual cells (not cell types) of a reasonably large set of different neurons in a systematic fashion. In my previous review I criticized that the methods used were not beyond the state of the art (which still holds), but in this more clear presentation of the differences to other studies the authors have valorized their dataset and modeling effort much better. Also, the authors now give better context to their work (such as the reference to the complementary effort of building detailed morphologically models from the same dataset) and integrate better with community resources.

My biggest problem is still the section on "Models and cell type characterization". In the response to my criticism to their approach they agree that indeed training classifiers on different datasets and then comparing the classifiers' performance can be misleading, but they argue that from the single model to experimental data comparison they have confidence that this is not the case. I am afraid this does not convince me; the classifiers' performances are too bad and too different to make any conclusions unless there is some indication that the individual classes are comparable. Minimally, I would expect to see a direct quantification of the mean + variability of features in the different classes which then could be compared between the same class across classifier/input combinations. Another test (at least for comparing experiment with model features) could be to train on experiments and use the same classifier on model features (without retraining). Both are easily done and can be added to the supplementary material if it supports the authors claims, if however the differences between two classes in different classifier/input combinations are too big, I fear the section may need to be reworked based on a more direct quantitative approach (rather than indirectly using classifier performance).

Reviewer #4 (Remarks to the Author):

I have reviewed the rebuttal and the revised manuscript and note that my comments have been fully and satisfactorily addressed.

Response to the Editor:

We thank the reviewers again for reviewing our re-submitted manuscript and are grateful that three of the four reviewers found that we fully addressed their concerns and comments. Reviewer 3 felt that we improved our manuscript but still had remaining concerns about our results relating to classification.

We have added three new supplementary figures to the manuscript in response to Reviewer 3's comments. We have presented a direct comparison of the experimental and model features by class in the first two figures, and we show the new classification results suggested by the reviewer in the third new figure. These new results are consistent with our conclusions in the previous version, and we believe they should address all the reviewer's remaining concerns as we understand them. Our specific responses to Reviewer 3's comments are found below.

Response to Reviewer 3:

The authors present a revised version of their manuscript, now entitled "Systematic generation of biophysically detailed models for diverse cortical neuron types". They have addressed multiple of the different reviewers' comments and overall substantially improved the manuscript.

Overall, the manuscript now describes much clearer what are the contributions of the study, in particular that they are fitting models for individual cells (not cell types) of a reasonably large set of different neurons in a systematic fashion. In my previous review I criticized that the methods used were not beyond the state of the art (which still holds), but in this more clear presentation of the differences to other studies the authors have valorized their dataset and modeling effort much better. Also, the authors now give better context to their work (such as the reference to the complementary effort of building detailed morphologically models from the same dataset) and integrate better with community resources.

My biggest problem is still the section on "Models and cell type characterization". In the response to my criticism to their approach they agree that indeed training classifiers on different datasets and then comparing the classifiers' performance can be misleading, but they argue that from the single model to experimental data comparison they have confidence that this is not the case. I am afraid this does not convince me; the classifiers' performances are too bad and too different to make any conclusions unless there is some indication that the individual classes are comparable. Minimally, I would expect to see a direct quantification of the mean + variability of features in the different classes which then could be compared between the same class across classifier/input combinations. Another test (at least for comparing experiment with model features) could be to train on experiments and use the same classifier on model features (without retraining). Both are

easily done and can be added to the supplementary material if it supports the authors claims, if however the differences between two classes in different classifier/input combinations are too big, I fear the section may need to be reworked based on a more direct quantitative approach (rather than indirectly using classifier performance).

In response to the reviewer's suggestion to present the distributions of the features in the different classes directly, we have added Supplementary Figs. 3 and 4 to our submission. These figures plot the feature distributions by transgenic line-based classes (Supplementary Fig. 3) and by classes from unsupervised clustering (Supplementary Fig. 4), and they directly compare the experimental features to the model features within the same class. Across the features and classes, we see quantitatively similar distributions in both experiments and models.

We have also followed the reviewer's suggestion of training a classifier based on experimental features, then testing it using the model features without re-training. The results are presented in the new Supplementary Fig. 5. We find that the classifiers (for both transgenic-based classes and classes from unsupervised clustering) perform well (~75% accuracy) when applied to the model features.

We appreciate the reviewer's request for these additional analyses, and we believe these new figures provide additional support to our conclusions from Fig. 7.

REVIEWERS' COMMENTS:

Reviewer #3 (Remarks to the Author):

I thank the authors for having addressed diligently my concerns regarding the comparison of experimental and model features. I have no further comments.